# ASPP proteins discriminate between PP1 catalytic subunits through their SH3 domain and the PP1 C-tail

M. Teresa Bertran[1], Stéphane Mouilleron [2], Yanxiang Zhou[1], Rakhi Bajaj[3], Federico Uliana[4], Ganesan Senthil Kumar[3], Audrey van Drogen[4], Rebecca Lee[2], Jennifer J. Banerjee[1], Simon Hauri[4], Nicola O'Reilly [5], Matthias Gstaiger[4], Rebecca Page[3], Wolfgang Peti[3] & Nicolas Tapon [1]

Serine/threonine phosphatases such as PP1 lack substrate specificity and associate with a large array of targeting subunits to achieve the requisite selectivity. The tumour suppressor ASPP (apoptosis-stimulating protein of p53) proteins associate with PP1 catalytic subunits and are implicated in multiple functions from transcriptional regulation to cell junction remodelling. Here we show that *Drosophila* ASPP is part of a multiprotein PP1 complex and that PP1 association is necessary for several in vivo functions of *Drosophila* ASPP. We solve the crystal structure of the human ASPP2/PP1 complex and show that ASPP2 recruits PP1 using both its canonical RVxF motif, which binds the PP1 catalytic domain, and its SH3 domain, which engages the PP1 C-terminal tail. The ASPP2 SH3 domain can discriminate between PP1 isoforms using an acidic specificity pocket in the n-Src domain, providing an exquisite mechanism where multiple motifs are used combinatorially to tune binding affinity to PP1.

[1] Apoptosis and Proliferation Control Laboratory, The Francis Crick Institute, 1 Midland Road, London NW1 1AT, UK. [2] Structural Biology - Science Technology Platform, The Francis Crick Institute, 1 Midland Road, London NW1 1AT, UK. [3] Chemistry and Biochemistry Department, University of Arizona, 1041 E. Lowell Street, Biosciences West, 517, Tucson, AZ 85721, USA. [4] Department of Biology, Institute of Molecular Systems Biology, ETH Zürich, 8093 Zürich, Switzerland. [5] Peptide Chemistry Laboratory, The Francis Crick Institute, 1 Midland Road, London NW1 1AT, UK. These authors contributed equally: M. Teresa Bertran, Stéphane Mouilleron, Yanxiang Zhou. Correspondence and requests for materials should be addressed to S.M. (email: stephane.mouilleron@crick. ac.uk) or to N.T. (email: nic.tapon@crick.ac.uk)

The large number of serine/threonine kinases (over 500 in humans) dwarfs the ~40 serine/threonine phosphatase catalytic subunits[1], posing a major challenge for the specific dephosphorylation of kinase substrates. For the most abundant of these phosphatase catalytic subunits, PP1 and PP2A (phosphoprotein phosphatase 1/2A), this challenge is met by a large collection of accessory subunits, which recruit specific substrates, prevent promiscuous dephosphorylation events, and/or tether phosphatases to discrete subcellular locations[1].

Over 200 PP1-interacting proteins (PIPs) have been identified to date[2–4], ~70% of which contain the small linear motif (SLiM) RVxF, which binds a hydrophobic groove in the PP1 catalytic subunit[5,6]. In addition to the RVxF motif, many other PP1-binding SLiMs have been identified, such as SILK[5,7–9], ΦΦ[10], and KiR (Ki67–RepoMan) motifs[11]. Several SLiMs are often combined within an intrinsically disordered domain to form a high affinity PIP/PP1 complex[2]. This is the case for several PIPs where the structural basis for PP1-binding has been elucidated[12], including the targeting subunits MYPT1 (Myosin phosphatase targeting subunit 1), which uses an RVxF and a MyPHONE motif to contact PP1[7,13] and Spinophilin (also known as Neurabin, RVxF and ΦΦ)[14]. A subset of these PIPs occludes (e.g. Spinophilin, PNUTS) or extends (e.g. MYPT1) some of the three PP1 substrate-binding grooves, thereby increasing catalytic subunit specificity[3]. Others promote specificity by recruiting PP1 catalytic subunits to their substrates or a particular subcellular localisation[2–4].

In mammals, four PP1 catalytic subunits encoded by three genes exist: the broadly expressed PP1α, PP1β and PP1γ$_1$, and the testis-specific PP1γ$_2$. Genetic analysis in model organisms has shown that different PP1 isoforms perform overlapping but distinct cellular functions. Complementation tests substituting the single Saccharomyces cerevisiae PP1 catalytic subunit GLC7 with different human PP1 isoforms showed that each human PP1 catalytic subunit only fulfils a subset of the functions of yeast PP1[15]. In Drosophila, PP1β overexpression cannot complement for loss of PP1β activity[16]. Furthermore, Dam-ID experiments indicate that human PP1α, PP1β and PP1γ$_1$ display distinct chromatin association profiles, suggesting they regulate non-overlapping gene sets[17].

Thus, isoform selectivity is thought to be a key feature of regulatory PIPs, several of which display isoform preferences, such as MYPT1[13], Spinophilin[18,19], RepoMan[11,20] and Ki67[11,21], though limited mechanistic information exists on how this is achieved[2–4,12,22]. In the case of Ki67 and RepoMan, two PIPs involved in mitotic exit, the KiR-SLiM motif determines preference toward PP1γ through a single amino acid change in a binding pocket in the catalytic domain of PP1[11]. Human PP1s differ mainly in their N- and, most notably, C-termini (see Fig. 4a[22]), suggesting that in many cases these could provide the basis for subunit selectivity. The PP1 termini are unstructured[23], therefore their involvement in PP1 recruitment by PIPs has been difficult to dissect molecularly. An exception is MYPT1, whose ankyrin repeats associate with amino acids 301–309 in the PP1 C-tail, and are proposed to drive selectivity towards PP1β[13], though the MyPHONE-binding region of PP1β may also participate in selectivity[24]. However, the extreme C-terminus (PP1α$^{309-330}$), which contains a type 2 SH3-binding motif (PPII – PxxPxR) and a variable C-tail, has never been crystallised and its contribution to PIP recruitment remains under-studied.

The ASPP (apoptosis-stimulating protein of p53) protein family, which in mammals is composed of ASPP1, ASPP2 and iASPP (inhibitor of ASPP), are RVxF-containing PIPs (RARL in the case of iASPP)[25]. p53BP2, a fragment of ASPP2, was one of the first RVxF-containing PP1 interactors to be identified[6,26], while Drosophila ASPP was recovered in a two-hybrid screen for

PIPs[27]. ASPP proteins have well characterized functions as modulators of gene transcription through the p53 family[28] and also regulate cell–cell contact remodelling in mammals and flies[29–31]. Indeed, mammalian ASPP2 and Drosophila ASPP localise at tight junctions and adherens junctions (AJs), respectively[29–31], and are required for junctional stability, at least in part by recruiting the polarity protein Par-3 (Bazooka in flies)[29,30,32], although the role of the ASPP/PP1 association has not been examined in this context.

The function of ASPP proteins in recruiting PP1 catalytic subunits has been examined in two contexts. First, during the cell cycle, ASPP1/2 and PP1α dephosphorylate the kinetochore component Hec1 to promote microtubule/kinetochore attachment[33], and c-Nap1 to induce centrosomal linker reassembly at the end of mitosis[34]. Second, the growth-promoting transcriptional coactivator targets of the Hippo pathway, YAP (yes-activated protein) and TAZ (transcriptional coactivator with PDZ-binding motif) are dephosphorylated on an inhibitory site by ASPP2/PP1, promoting their transcriptional activity[35,36]. As well as their PP1-binding RVxF/RARL motifs, ASPP proteins have C-terminal ankyrin repeats (like MYPT1) followed by an SH3 domain (Supplementary Fig. 1a). Mutation of the PP1 C-terminal PPII compromises binding to ASPP proteins[25], while an SH3 domain mutation in iASPP impairs PP1 association[37], suggesting that ASPP family members can engage the PP1 C-tail. Here, we characterize the ASPP/PP1 complex structurally and functionally. Our data show that ASPP proteins use their SH3 domain to discriminate between different PP1 isoforms based on their C-tail and provide structural insights into the function of the PP1 C-tail.

## Results

**Characterisation of the Drosophila ASPP/PP1 complex.** Our previous affinity purification-mass spectrometry (AP-MS) data[38] suggested that ASPP1/2 are part of a tightly interlinked protein network comprising PP1 catalytic subunits, as well as the PP1 inhibitor IPP2 (Fig. 1a). In addition, this putative complex contains N-terminal RASSF (ras-association domain family) (RASSF7-10) scaffold proteins[39] and different isoforms of the small coiled-coil protein CCDC85. Prior to performing a genetic analysis of this complex, we first sought to biochemically characterise the Drosophila ASPP/PP1 complex. First, we verified the binary interaction between ASPP and the four Drosophila PP1 catalytic subunits by co-immunoprecipitation (co-IP) from Drosophila S2 cell lysates (Fig. 1b). As expected, ASPP associated with PP1 catalytic subunits, though it displayed a marked preference for PP1α96A and PP1β9C, with PP1α87B and PP1α13C weakly associating with ASPP. Like mammalian ASPP1 and ASPP2, Drosophila ASPP possesses a PP1-binding RVxF motif at its C-terminus, preceding the ankyrin repeats. We mutated the ASPP RVxF motifs (RVSF to RASA; ASPP$^{FA}$, see Supplementary Fig. 1a). Association of ASPP$^{FA}$ to the PP1 catalytic subunits was substantially reduced in co-IP experiments compared with ASPP$^{wt}$ (Fig. 1b), suggesting the RVxF is required for the Drosophila ASPP/PP1 interaction, as is the case for mammalian ASPP1/2[33,35].

We previously showed that the Drosophila ASPP and RASSF8 mutants share many phenotypic similarities, suggesting common biological functions[31], and our Mass Spectrometry analysis suggested that mammalian RASSF7 and RASSF8 (the Drosophila RASSF8 orthologs) are associated with ASPP/PP1 complexes in HEK293 cells[38]. Interestingly, in S2 cells, which do not express endogenous ASPP or RASSF8[31], PP1α96A associated with RASSF8 only in the presence of exogenous ASPP (Fig. 1c), suggesting that RASSF8 indirectly associates with PP1 via ASPP.

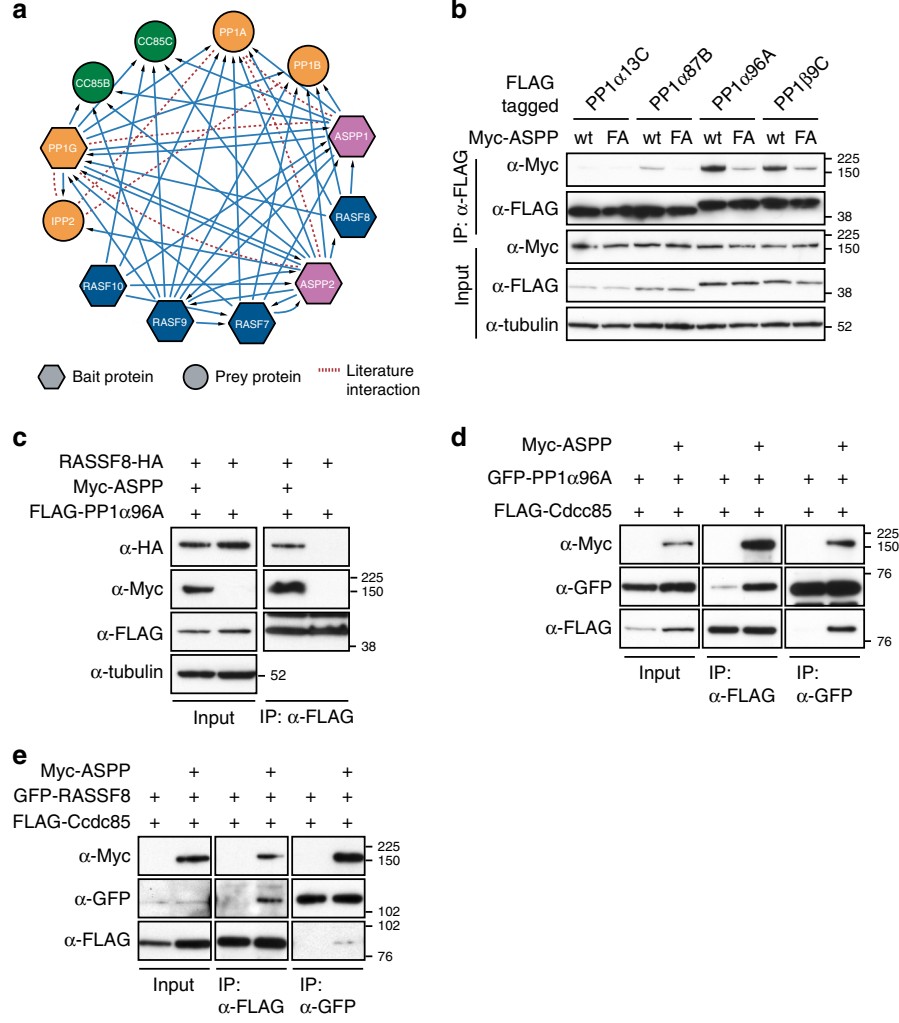

**Fig. 1** Biochemical characterisation of the ASPP/PP1 complex. **a** Summary of protein-protein interactions in the ASPP/PP1 complex detected by AP-MS from ref. [38] (solid blue lines) and collated from the literature (dashed red lines). The baits used in ref. [38] are shown as hexagons, preys are shown as circles. **b–e** Western blots of co-IP experiments using lysates of transfected S2 cells, probed with indicated antibodies. **b** Mutation of V812 and F814 to alanine within the RVxF motif of ASPP (ASPP[FA]) decreases the binding between PP1 isoforms and ASPP). **c** RASSF8 binds to PP1α96A via ASPP. RASSF8 and PP1α96A co-IP with each other only when ASPP is present. **d** Increased PP1α96A/Cdcc85 co-IP in the presence of ASPP. **e** RASSF8 and Ccdc85 co-IP with each other only when ASPP is present

Next, we examined Ccdc85, which was also associated with PP1, ASPP1/2 and RASSF7-10 in HEK293 cells[33,38]. In the absence of ASPP, FLAG-Ccdc85 IPs recovered a small amount of PP1α96A, which was strongly increased upon ASPP co-transfection (Fig. 1d). Conversely GFP-PP1α96A IP recovered Ccdc85 only in the presence of ASPP (Fig. 1d). Thus, Ccdc85 can interact with PP1 through ASPP, though a low affinity direct interaction is possible.

Since RASSF8 and Ccdc85 could form a complex with ASPP/PP1, we tested whether these might represent separate complexes or if a tetrameric complex was possible. First, we mapped the domain of ASPP involved in the RASSF8 and Ccdc85 interactions by performing co-IPs with three truncations: 1–234 (comprising the coiled-coiled region), 231–795 (the divergent middle region) and 796–1020 (comprising the RVxF motif, the ankyrin repeats and SH3 domain) (Supplementary Fig. 1a). These experiments revealed that the ASPP coiled-coil domain is required for the recruitment of both RASSF8 and Ccdc85 to the ASPP/PP1 complex in *Drosophila* S2 cells (Supplementary Fig. 1b, c). Mutating the PP1-binding RVxF motif at the ASPP C-terminus had no effect on the ASPP interaction with RASSF8 or Ccdc85

(Supplementary Fig. 1d, e). Reciprocal co-IP of RASSF8 with Ccdc85 was only possible in the presence of ASPP (Fig. 1e). Thus, our biochemical data indicate that an ASPP/PP1 complex also comprising Ccdc85 and RASSF8 can exist in cultured S2 cells. In Human Embryonic Kidney (HEK) 293T cells, we were able to IP RASSF8 and endogenous PP1α with ASPP2, confirming the Mass Spectrometry and S2 cell results (Supplementary Fig. 1f).

**ccdc85 mutants phenocopy the ASPP eye phenotypes.** We had previously shown that *ASPP* and *RASSF8* mutants share many phenotypic similarities[31]. In particular, these mutants display a rough eye phenotype due to an excess number and aberrant organisation of the inter-ommatidial cells (IOCs), which separate each unit of the *Drosophila* compound eye[31] (see Fig. 2a for a schematic of wild type ommatidial organization). Since Ccdc85 is associated with the ASPP/PP1 complex in cell culture, we wished to analyse the effects of its genetic inactivation. We generated a *ccdc85* mutant (*ccdc85[C1.1]*) by transposon mobilisation (Supplementary Fig. 2a–c, see the Methods section). *ccdc85[C1.1]* animals were viable but displayed a rough eye phenotype similar to that of

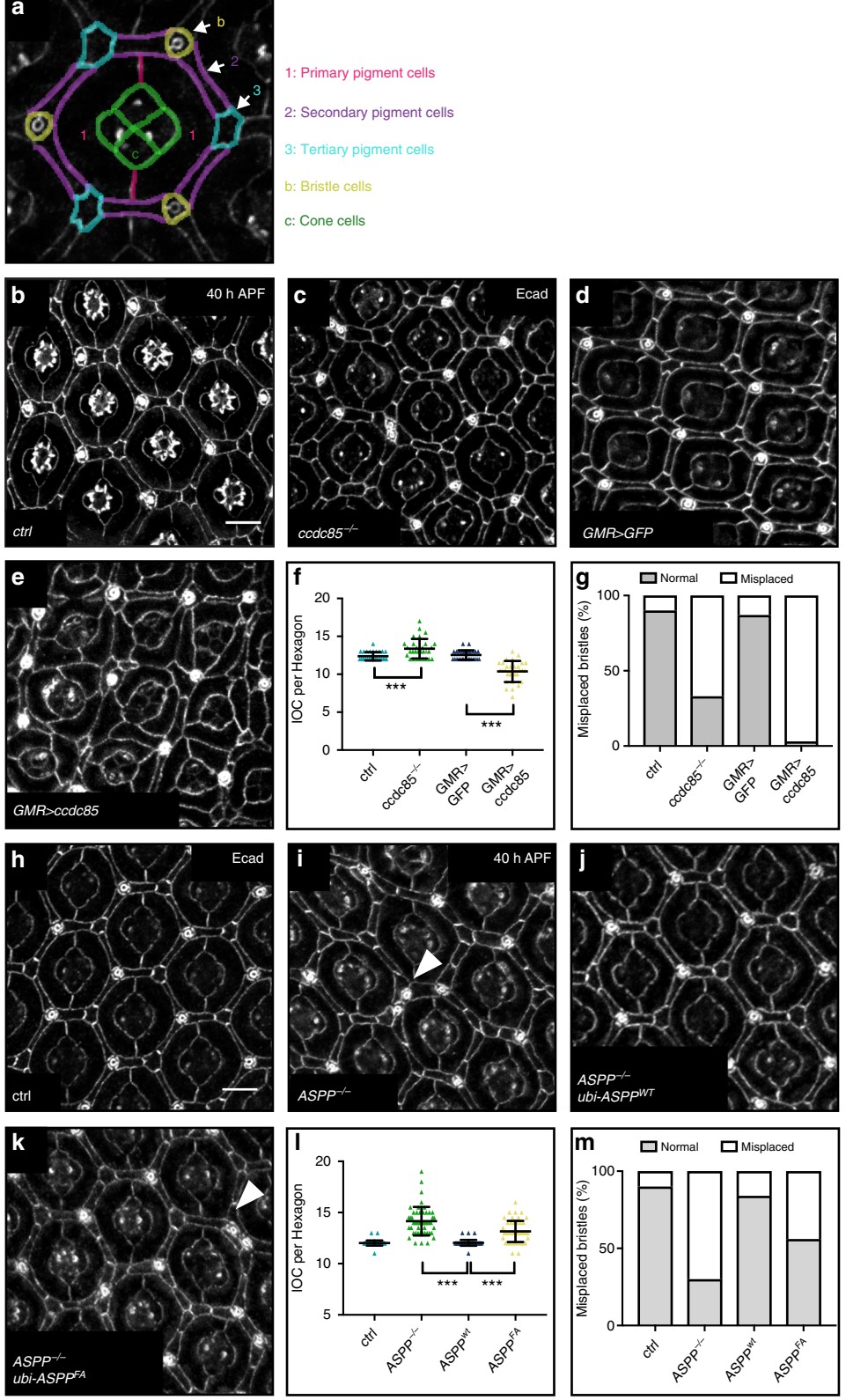

1: Primary pigment cells
2: Secondary pigment cells
3: Tertiary pigment cells
b: Bristle cells
c: Cone cells

*ASPP* and *RASSF8* mutants, either as homozygotes or in *trans* to a deficiency (*Df(2L)Exel7014*) or the original *P{XP}d06579* transposon insertion.

We quantified IOC numbers and bristle placement in *ccdc85C1.1* pupal retinas at 40 h APF (after puparium formation) stained with an anti-E-cadherin (E-cad) antibody to visualise cell boundaries (Fig. 2). Like *ASPP* and *RASSF8* mutants (Fig. 2i, l and

ref. [31]), *ccdc85C1.1* mutants displayed increased IOC numbers and a substantial number of bristle displacements (Fig. 2b, c, f, g) though the IOC increase was milder than *ASPP* mutants (*ccdc85*: 13.37 vs *ASPP*: 14.15, compared with 12.37 for control retinas). Conversely, Ccdc85 overexpression under the control of the GAL/ UAS system using the eye-specific *GMR-GAL4* driver reduced IOC numbers and also induced bristle displacements (Fig. 2d–g).

**Fig. 2** ccdc85 and the RVxF motif of ASPP are required for eye patterning. **a** Schematics of a wild type ommatida showing normal number and arrangement of (1) primary, (2) secondary and (3) tertiary pigment, (b) bristle and (c) cone cells. **b–e**, **h–k** Confocal X–Y sections of pupal retinas of the indicated genotypes at 40 h APF stained with anti-E-cadherin antibody to mark cell outlines. Misplaced bristles (arrowheads) are indicated. See Supplementary table 1 for full genotypes. Scale bar = 10 μm. **f** Quantification of IOCs per ommatidial unit for the indicated genotypes. ccdc85 null mutants have an increased number of IOCs (13.37 ± 1.3), flies misexpressing Ccdc85 have a decreased number of IOCs (10.53 ± 1.4). Error bars represent the standard deviation ($n = 30$ hexagons from six retinas). *** indicates $p < 0.001$ using unpaired Student's $t$-tests. **g** Loss of ccdc85, or Ccdc85 overexpression cause bristle misplacements. Bristle clustering or placement of a bristle in a neighbouring position that should be occupied by a tertiary pigment cell were counted as misplacements ($n = 30$ from six retinas). **l** Quantification of IOC per ommatidial units for the indicated genotypes. ASPP but not ASPP-FA expression restores the number of IOCs in ASPP mutants (12.02 ± 0.29 vs 13.15 ± 1.04 compared to 14.15 ± 1.39 in ASPP mutants). Error bars represent the standard deviation ($n = 50$ hexagons from six retinas). A one-way ANOVA test was carried out to determine if the differences among means were significantly different from each other. Three pairwise comparisons were carried out (ASPP⁻/⁻ vs ubi-ASPPwt, ubi-ASPPwt vs ubi-ASPPFA and control vs ubi-ASPPwt) and $p$-values were adjusted using a Bonferroni correction. Significant differences are marked. *** indicates $p < 0.001$. **m** Quantification of the bristle misplacements in (**g–j**)

Thus, ccdc85 disruption elicits a similar eye phenotype to ASPP and RASSF8 loss, while its overexpression has the opposite effect, suggesting that these three binding partners share common functions in vivo.

**Loss of the RVxF motif disrupts ASPP function in vivo**. Although the ASPP and ccdc85 mutants share phenotypic similarities, this does not prove that their shared developmental function pertains to PP1 regulation. To address this question, we tested the effect of disrupting the ASPP/PP1 interaction in vivo. We generated an ASPP rescue construct where the ASPP cDNA is expressed at low levels under the control of the ubiquitin 63E promoter with a N-terminal GFP tag (ubi > ASPP, see experimental procedures). Expression of this construct in ASPP mutant animals fully rescued both the increased IOC number (Fig. 2h–j and l) and bristle displacement defects (Fig. 2h–j and m). In contrast, expression of ASPPFA inserted at the same locus only partially rescued both defects (Fig. 2h, k–m). Mosaic expression of the ubi > ASPPwt and ubi > ASPPFA constructs using the Flp/FRT system indicated that both constructs are normally localised at the AJs of epithelial cells, and expressed at similar levels (Supplementary Fig. 2d–i).

Next, we examined the effect of the ASPPFA mutation on other ASPP phenotypes[31]. While ubi > ASPPwt fully rescued the wing size increase (Fig. 3a, b, d and e) and wing notching upon single copy loss of csk (C-terminal src kinase, Fig. 3f, h), ubi > ASPPFA only partially rescued both phenotypes (Fig. 3c, d, e, g and h). ASPP mutant animals frequently display a duplication of the anterior scutellar macrochaete of the adult thorax (Supplementary Fig. 3a, b). This defect is partially rescued by ubi > ASPPwt and not at all by ubi > ASPPFA. Finally, opposite to the loss-of-function (Fig. 3e), overexpression of ASPP using the GAL4/UAS system under the control of the wing-specific MS1096-GAL4 driver reduced wing size by ~10% (Fig. 3i, j, l and m), while ASPPFA expression had little effect (Fig. 3k, l and m). Thus, association with PP1 is required for several in vivo functions of ASPP.

**ASPP association with PP1 subunits requires multiple motifs**. An interesting characteristic of the different PP1 catalytic subunits in Drosophila and mammals is the diversity of their C-terminal tails (hereafter referred to as C-tail) (Fig. 4a). In particular, two of the Drosophila PP1s (PP1α87B and PP1α13C) have truncated C-tails (Fig. 4a) and show a markedly reduced association with ASPP in co-IP experiments (Fig. 1b) compared with PP1α96A and PP1β9C, which have an extended C-tail, like all the mammalian PP1c isoforms (Fig. 4a). We generated C-terminally truncated versions of PP1α96A and PP1β9C lacking the C-tail and observed that these displayed a strongly reduced ability to

associate with ASPP in co-IP experiments (Fig. 4b). This suggests that ASPP can discriminate between different PP1 isoforms based on the PP1 C-tail.

Like mammalian ASPP1/2, Drosophila ASPP has an SH3 domain at its C-terminus (Supplementary Fig. 1a), while the PP1 C-tail of PP1α96A and PP1β9C and all the mammalian PP1 isoforms contain a highly conserved PPII (ref. [25] and Fig. 4a). Indeed, recent work showed that deletion or mutation of the PP1 C-tail compromises its ability to bind ASPP1, ASPP2 and iASPP[25]. We generated a mutant ASPP construct with a tryptophan to lysine mutation in the hydrophobic patch of the SH3 domains (W897K - ASPPWK), which abolishes ligand binding while preserving SH3 domain structure[40]. ASPPWK had reduced binding to PP1α96A and PP1β9C in co-IP experiments (Fig. 4c). In contrast, ASPPWK binding to PP1α87B and PP1α13C was not affected, since these lack the C-terminal SH3-binding motif (Fig. 4c). Mutation of both the RVxF and the SH3 domains (ASPPFA-WK) further reduced the binding between ASPP and PP1α96A to background levels (Fig. 4d). Thus, both the RVxF and SH3 domains of ASPP participate in PP1 docking. The ASPP/PP1 interaction is an example of a PIP that engages the PPII motif of the PP1 C-tail. In order to structurally characterize this interaction modality, we co-crystalized human ASPP2 with PP1α.

**Crystal structure of ASPP2:PP1α**. We solved the crystal structure of human PP1α⁷⁻³³⁰ bound to ASPP2⁹²⁰⁻¹¹²⁰ (hereafter referred to as PP1α:ASPP2 complex) at a resolution of 2.15 Å (Table 1). The overall structure of the PP1 catalytic domain is very similar to other reported PP1 structures[12]. ASPP2 binds to PP1α through three surfaces of interaction, covering ~1200 Å² of solvent-accessible area: the RVxF motif, the first ankyrin motif and the SH3 domain, respectively, burying 530, 120 and 500 Å² of solvent-accessible area (Fig. 5a, Supplementary Fig. 4a). ASPP2 residues ⁹²¹RVKF⁹²⁴, constitute the canonical RVxF motif, which binds PP1 hydrophobic pocket in an extended conformation similar to what has been observed previously for other PP1-binding proteins (Fig. 5b)[8,10,11,13,14,41–43]. The second patch of interactions is formed by four acidic residues from the ASPP2 first ankyrin motif facing PP1α^K260 and PP1α^R261 (Fig. 5c). PP1α^R261 forms a network of hydrogen bonds with ASPP2^D932 and ASPP2^E938. This observation is in agreement with previous work showing that PP1α^R261S mutation reduced ASPP2 binding to PP1α[25]. A similar interaction has been observed in the structure of PP1β bound to MYPT1[13].

The electron density observed for the PP1α:ASPP2 structure allowed the modelling of the seven conserved residues from its PPII motif (PP1α³¹⁷RPITPPR³²³) bound to ASPP2 SH3 domain and forming the third surface of interaction (Fig. 5d, e and Supplementary Fig. 4b). Native mass spectrometry analysis

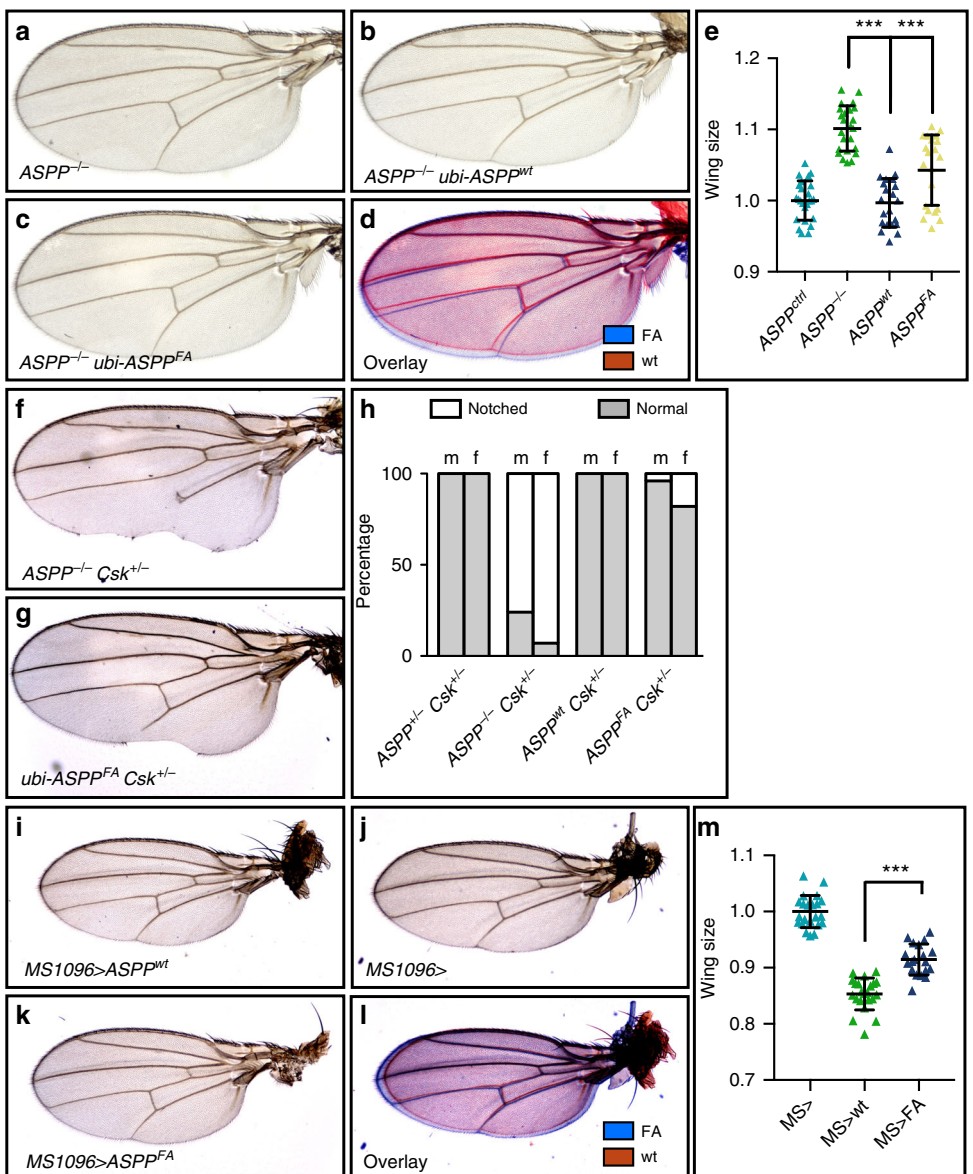

**Fig. 3** The RVxF motif of ASPP is required for wing development. **a–d** Wings from adult female flies. *ASPP* null mutants have enlarged wings (**a**). The increase in wing size is rescued by the expression of ASPP (**b**), but only partially by *ASPP^FA* (**c**). **d** Overlay of (**b**) and (**c**). **e** Quantification of wing sizes normalised to control wing size. A one-way ANOVA test indicated that the differences among means were significant ($p < 0.001$). Error bars represent the standard deviation. Two pairwise comparisons were carried out (*ASPP^{−/−}* vs *ubi-ASPP^{wt}* and *ubi-ASPP^{wt}* vs *ubi-ASPP^{FA}*) and *p*-values were adjusted using a Bonferroni correction. Significant differences are marked. *** represents $p < 0.001$. At least 20 wings were counted per genotype. **f**, **g** Wings from adult female flies. Loss of one copy of *csk* in an *ASPP* null background leads to wing notching (**f**). Expressing exogenous ASPP^{FA} does not fully rescue the notching phenotype (**g**). **h** Quantification of wing notching phenotype. Only the genotypes shown in (**f**) and (**g**) had visible notching. 100 wings were analysed per genotype. **i–l** Wings from adult male flies. Compared to expressing ASPP^{FA} with the *MS1096-GAL4* driver (**k**), ASPP expressing wings (**k**) are smaller. Control wings (**i**) are of similar size to ASPP-FA wings. **l** Overlay of (**j**) and (**k**). **m** Quantification of wing sizes normalised to control wing size. Error bars represent the standard deviation. An unpaired *t*-test showed that mean wing size in *MS1096 > ASPP* and *MS1096 > ASPP^{FA}* differ significantly from each other. *** represents $p < 0.001$. At least 20 wings were counted per genotype

indicated that the PP1:ASPP2 complex shows no evidence of degradation after 7 days at 20 °C, therefore the C-tail residues not observed in the electron density are most likely flexible rather than proteolytically removed (Figure Supplementary Fig. 4c). PP1α^{317–323} adopts the characteristic right handed type II polyproline helix containing three residues per turn[44]. The two conserved prolines PP1α^{P318} and PP1α^{P321} sit in two xP binding grooves formed by ASPP2^{L1113/W1066} and ASPP2^{P1110/W1097}, respectively (Fig. 5a, d, e). PP1α^{R323} is tightly coordinated by three acidic residues from the ASPP2 SH3 domain: ASPP2^{D1074}

and ASPP2^{E1075} from the RT loop and ASPP2^{E1094} from the n-Src loop.

**ASPP2's SH3 acidic specificity pocket promotes PP1α binding.** The affinity of PP1α^{7-330}:ASPP2^{920-1120} was measured by Iso-thermal Calorimetry (ITC) and by Bio Layer Interferometry (BLI). PP1α was found to bind tightly to ASPP2 with an affinity of 6.1 and 13.6 nM by ITC and BLI methods, respectively (Fig. 6a, b). Truncation of the PP1 C-tail (PP1α^{7-300}) has a dramatic effect, with no binding detected by ITC (Fig. 6a). To gain a better

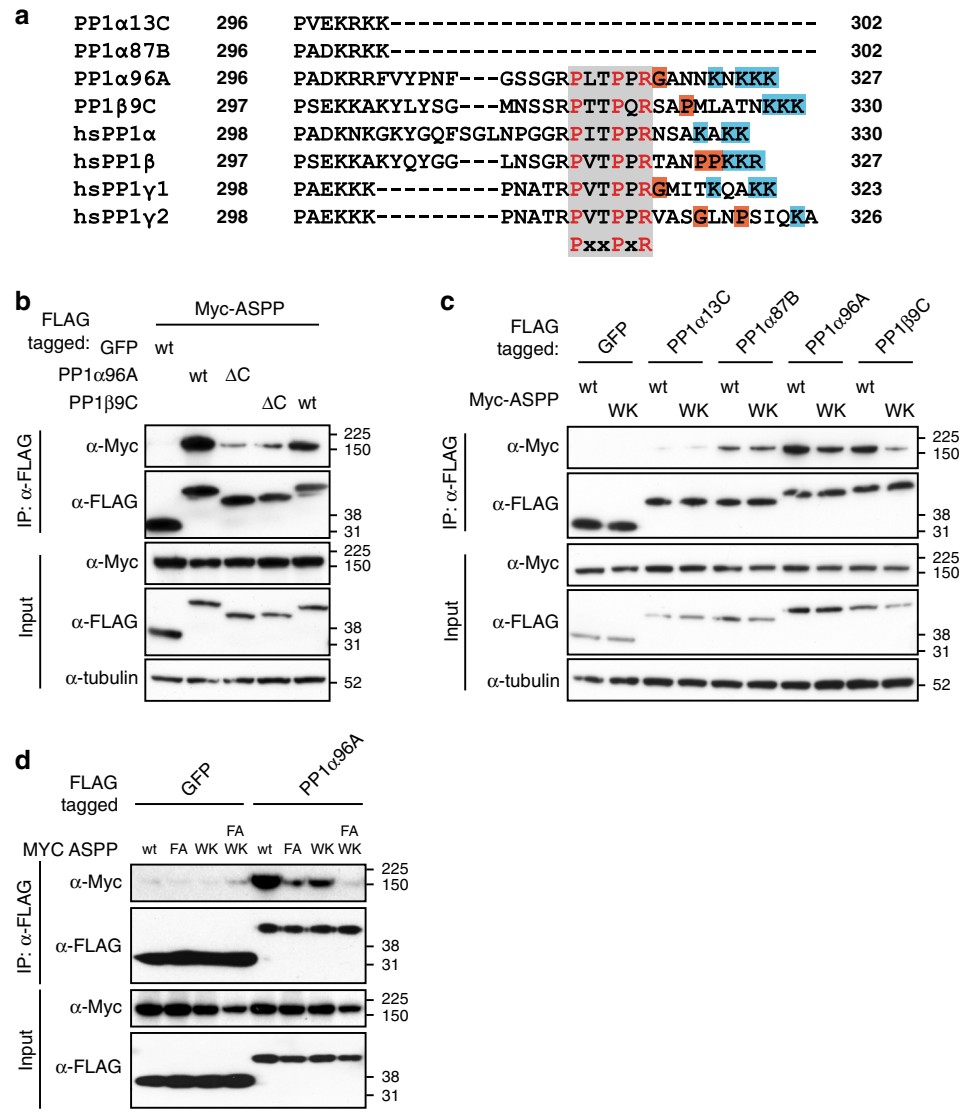

**Fig. 4** The SH3 domain of ASPP binds to the PP1 C-tail. **a** Alignment of the C-termini of human and *Drosophila* PP1s. PP1α13C and PP1α87B have shorter C-termini that lack the PxxPxR motif (class II SH3 domain binding motif). All human PP1 isoforms possess this motif. The PxxPxR motif is highlighted in grey. Positively charged residues after the PPII are highlighted in blue. **b, c** Western blots of co-IP experiments from lysates of transfected S2 cells, probed with indicated antibodies. **b** PP1α96A and PP1β9C need their C-termini for efficient binding to ASPP. PP1α96A-ΔC lacks residues 304–327 and PP1α9C-βC lacks residues 304–330. **c** The SH3 domain of ASPP is required for binding to PP1α96A and PP1β9C. The W987K mutation in the ASPP SH3 domain (ASPP^WK) reduces binding to PP1α96 A and PP1β9C but not PP1α13C and PP1α87B, which do not have the class II SH3 domain binding motif. **d** Western blots of co-IP experiments from lysates of transfected S2 cells, probed with indicated antibodies

understanding of the role of the PP1 C-tail in PP1:ASPP2 complex formation, we measured the binding affinity of various peptides corresponding to PP1 C-tails to ASPP2^920-1120 by ITC (Fig. 6c, Supplementary Fig. 5). The PP1α^301-330 C-tail binds ASPP2 with a surprisingly high affinity of 143 nM (Fig. 6c, Supplementary Fig. 5a). Mutation of PP1α^R323 to alanine completely abolished ASPP2 binding, highlighting the critical role of PP1α^R323 for ASPP2 binding (Fig. 6c, Supplementary Fig. 5b). The expected binding affinity of a PPII motif for its SH3 domain is in the 5–20 μM range[45,46]. The fact that ASPP2^920-1120 binds to the PP1α^301-330 C-tail almost two orders of magnitude above this value prompted us to consider the possibility that additional contacts not observed in the crystal structure may explain this high affinity.

The core PPII motif sequence PP1α^317-323 found in all PP1 C-tails binds ASPP2 with a much lower affinity of 5650 nM, whereas the same sequence extended by seven residues at its C-terminal

extremity (PP1α^317-330) binds with a much stronger affinity of 270 nM, comparable with PP1α^301-330 (Fig. 6c, Supplementary Fig. 5c–d). Thus, the C-terminal residues following the PPII motif are critical for ASPP2 binding. A similar observation has previously been reported for a few other PP motifs displaying unusually strong affinities for their respective SH3 domain[47–51]. In those complexes, the increased affinity arises from additional interactions between residues flanking the core PP sequence and the SH3 domain specificity pocket constituted by the variable region of the RT and n-Src loops[52]. A careful inspection of the electrostatic potential surface of ASPP2:PP1α structure reveals that ASPP2 SH3 domain contains a very acidic specificity pocket, formed by ^1090EDEDE^1094 from the n-Src loop and ^1073DDE^1075 from the RT loop (Fig. 5e, Supplementary Fig. 4d). We therefore hypothesized that this specificity pocket could accommodate the basic PP1α C-terminal residues ^324NSAKAKK^330 through a number of electrostatic interactions tethering the PP1α C-tail on

**Table 1 Data collection and refinement statistics (molecular replacement)**

|  | PP1:ASPP2 |
| --- | --- |
| *Data collection* | |
| Space group | P1 |
| Cell dimensions | |
| $a$, $b$, $c$ (Å) | 46.8, 81.6, 87.9 |
| $\alpha$, $\beta$, $\gamma$ (°) | 91.0, 91.8, 103.9 |
| Resolution (Å) | 59–2.15 (2.20–2.15)[a] |
| $R_{sym}$ or $R_{merge}$ | 0.28 (1.0) |
| $I$ /$\sigma I$ | 4.3 (1.5) |
| Completeness (%) | 99.8 (98.1) |
| Redundancy | 6.9 (5.7) |
| *Refinement* | |
| Resolution (Å) | 50–2.15 (2.18–2.15) |
| No. reflections | 68,468 |
| $R_{work}$/$R_{free}$ | 17.7/21.4 |
| No. atoms | |
| Protein | 7860 |
| Ligand/ion | 154 |
| Water | 245 |
| *B-factors* (Å$^2$) | |
| PP1 | 41.7 |
| ASPP | 59.7 |
| Ligand/ion | 60.5 |
| Water | 50.3 |
| R.m.s. deviations | |
| Bond lengths (Å) | 0.06 |
| Bond angles (°) | 0.74 |

[a]One crystal was used for data collection. Values in parentheses are for highest-resolution shell

the ASPP2 SH3 domain specificity pocket (Fig. 5e). Alanine substitution of the three C-terminal lysine residues PP1α$^{K327}$, PP1α$^{K329}$ and PP1α$^{K330}$ (hereafter referred to as PP1α$^{301-330}$ 3KA mutation) reduces the affinity for ASPP2 by 13-fold, down to 1900 nM (Fig. 6c, Supplementary Fig. 5e). These data strongly support a critical role of PP1α$^{K327}$, PP1α$^{K329}$ and PP1α$^{K330}$ in providing additional affinity and specificity to PP1α C-tail towards ASPP2 SH3 domain. Individual alanine mutations of these lysine residues show that they all contribute to ASPP2 binding (Fig. 6c, Supplementary Fig. 5f–h). However, mutation of PP1α$^{K327}$ to alanine showed the strongest effect with a fourfold decrease in affinity (Fig. 6c, Supplementary Fig. 5e).

**PP1 C-tails determine ASPP2 isoform preference**. All human PP1 isoforms share a high sequence similarity (92.5–100%) within their catalytic domains and differ mainly in their N-terminal and C-terminal extremities. They all contain a highly variable C-tail except for their conserved PPII motif (Fig. 4a). To evaluate the PP1 isoform preference of ASPP family members, we measured the affinity of human iASPP, ASPP1 and ASPP2 for each full-length PP1 isoform by BLI. ASPP1 did not display any PP1 isoform selectivity in vitro (Fig. 6d, Supplementary Fig. 6). iASPP showed a modest (twofold) preference for PP1α and PP1β versus PP1γ1 (Fig. 6d, Supplementary Fig. 6). In contrast, ASPP2 showed a strong preference for PP1α with an affinity of 13.6 nM which is 13- and 14-fold higher than that observed for PP1β and PP1γ1, respectively (Fig. 6d, Supplementary Fig. 6).

To test whether ASPP2 PP1 isoform preference is a consequence of the high sequence variability observed within the PP1 C-tail, we measured the affinity of each PP1 C-tail for ASPP2 by ITC. PP1α$^{301-330}$ binds ASPP2 with an affinity of 143 nM, which

is, respectively, 7, 5 and 15 times greater than the affinity measured for PP1β$^{297-327}$, PP1γ1$^{298-323}$ and PP1γ2$^{297-324}$ C-tails (Fig. 6c, Supplementary Fig. 5i–k). The differential affinities between ASPP2 and the different PP1 C-tails measured by ITC reflect those observed between ASPP2 and the full-length PP1 isoforms by BLI with PP1α binding over 13-fold stronger compared to PP1β and PP1γ1 (Fig. 6c, d). The chimeras generated by substituting the PP1β$^{297-327}$, PP1γ1$^{298-323}$ and PP1γ2$^{297-324}$ sequences C-terminal to the PPII motif by the seven C-terminal residues of PP1α (NSAKAKK) show elevated affinity to ASPP2, comparable to PP1α$^{301-330}$ (Fig. 6c, Supplementary Fig. 5l–m). Therefore, the reduced affinity observed for PP1β$^{297-327}$, PP1γ1$^{298-323}$ and PP1γ2$^{297-324}$ is the result of the sequence variability observed in their C-tails (Fig. 4a). All PP1 isoforms C-tails contain three basic residues within their last five C-terminal residues (except for PP1γ2 which has only one). However, the distance of the linker between the PPII motifs and those basic residues vary from 3 to 10 amino acid with the shorter one observed in PP1α. In addition, the longer PP1β, PP1γ1 and PP1γ2 linkers contain some glycine and proline residues which might influence the geometry of the C-tail (Fig. 4a). Together, these data show that ASPP2 discriminates between PP1 isoforms based on the PP1 C-tail, and more specifically on the basic residues following the PPII motif.

**ASPP2 specificity for PP1α resides in its n-Src loop**. Although ASPP1, ASPP2 and iASPP share high sequence homology in their C-terminal region, only ASPP2 displays selectivity for PP1α (Fig. 6d). Interestingly, we noticed some sequence differences within the variable n-Src loop (Supplementary Fig. 7a). For clarity, we numbered those n-Src loop residues from 1 to 5 (Supplementary Fig. 7a). The most acidic n-Src loop is found in ASPP2, while the other ASPPs lack some of the acidic residues. The n-Src loop of human ASPP1 and *Drosophila* ASPP are both missing acidic residues in positions 1 and 4, whereas human and *C. elegans* iASPP are both missing two acidic residues in positions 2 and 3, with the latter also lacking one in position 5. ASPP1 does not show specificity for PP1α (Fig. 6d), but mutation of the n-Src loop residues 1 and 4 in ASPP1 to the equivalent acidic ASPP2 residues (ASPP1$^{K1052E/S1055D}$) imparts specificity towards PP1α (Fig. 6d). Conversely, the single and double ASPP2 mutants mimicking the ASPP1 n-Src loop (ASPP2$^{E1090K/D1093S}$ and ASPP2$^{D1093S}$) display a drop in specificity for PP1α (Fig. 6d). These data highlight the critical role of the n-Src loop acidic specificity pocket in modulating ASPP1/2 selectivity towards PP1α.

The binding kinetics of human iASPP to all PP1 isoforms are very different compared to those observed for ASPP1/2 (Supplementary Fig. 6). iASPP binds PP1α with a lower apparent $K_D$ (2.6 nM instead of 13.6 nM for ASPP2), however, it shows a sevenfold lower $K_{on}$ and a 110-fold lower $K_{off}$ for PP1α compared to ASPP2. Therefore, once assembled, the PP1α:iASPP complex is more stable, with a half-life of 58 min versus 31 s for PP1α:ASPP2 complex. In order to understand how iASPP interacts with PP1α C-tails, we performed ITC measurement, showing that, though human iASPP binds the PP1α C-tail with relatively high affinity (395 nM), the binding is only modestly affected by the PP1α 3KA mutation (690 nM) (Fig. 6e, Supplementary Fig. 5n–o), showing that the iASPP/PP1 interaction is less reliant than ASPP2 on the PP1α C-terminal lysines and therefore likely involves other interactions. In support of this view, the iASPP SH3 domain specificity pocket is less acidic than that of ASPP2 (Supplementary Fig. 7a).

Interestingly, the double mutation ASPP2$^{D1091G/E1092P}$ in positions 2 and 3 of the ASPP2 n-Src loop mimicking the human

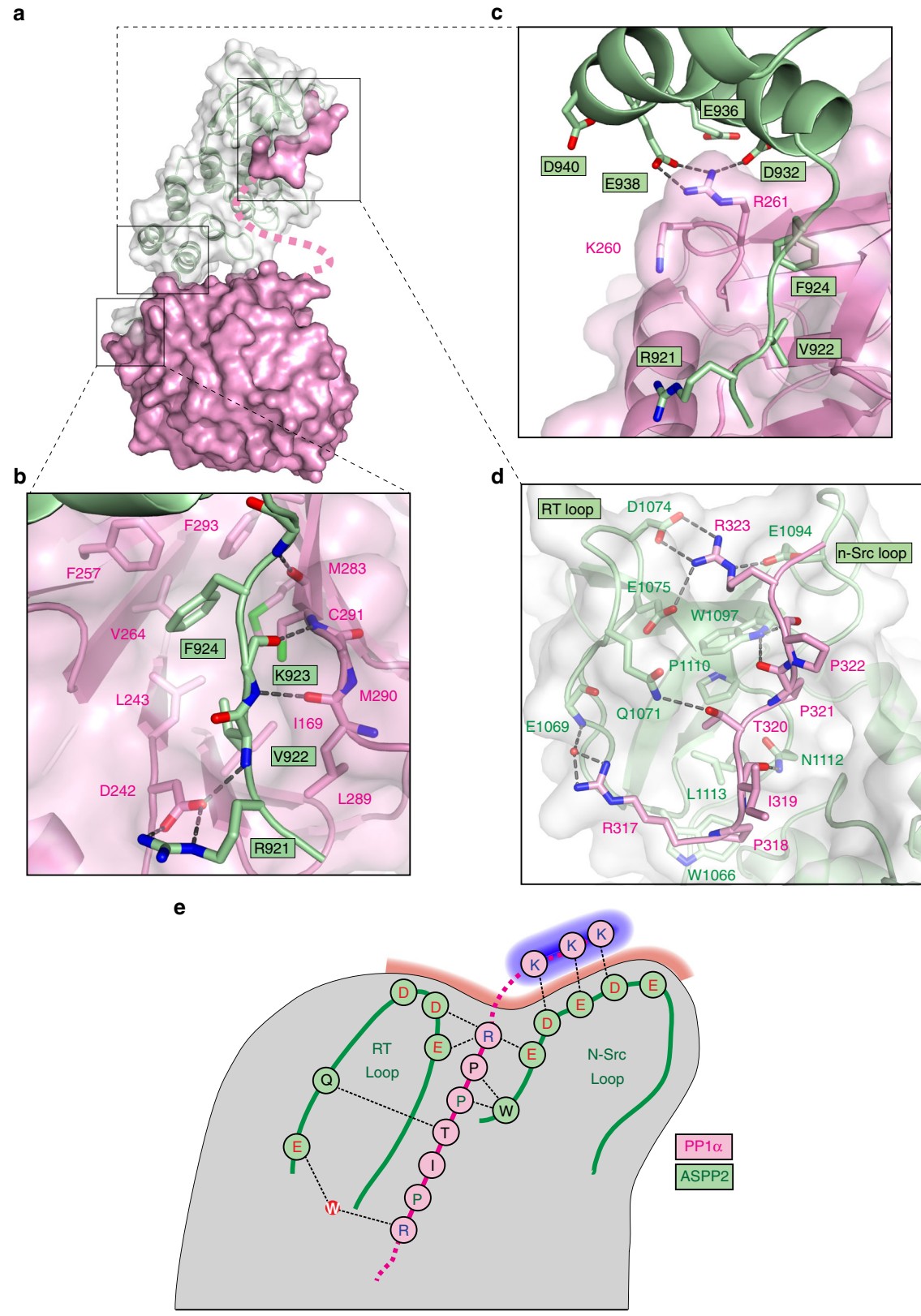

iASPP had little effect on binding to the PP1α C-tail by ITC or binding of the PP1α full-length by BLI (Fig. 6d–e, Supplementary Fig. 5p, Supplementary Fig. 6). However, the triple mutant ASPP2^D1091K/E1092V/E1094K mimicking the *C. elegans* iASPP n-Src loop with two charges inversion in positions 2 and 5 had a dramatic effect and abolished ASPP2 binding to the PP1α C-tail by ITC and reduced the affinity for full-length PP1α by almost 600-fold from 13.6 nM to 8.2 μM by BLI (Fig. 6d, e, Supplementary Fig. 5q, Supplementary Fig. 6). Thus, the PP1α C-tail lysines are recognised by the ASPP2 SH3 domain, while the ASPP1 and iASPP are less able to discriminate between PP1 isoforms because of their less acidic SH3 domain specificity pockets.

**Fig. 5** Crystal structure of PP1α:ASPP2 complex. **a** Crystal structure of PP1α:ASPP2 complex. ASPP2 is drawn in green cartoons and white transparent surface. PP1 is displayed in pink surface. **b** The ASPP2 RVxF motif is bound to the PP1 hydrophobic pocket: (PP1α residues I169, L243, F257, R261, V264, M283, L289, C291, F293). The interaction is stabilised by three backbone hydrogen bonds between ASPP2 residues $^{923}$KFN$^{925}$ and PP1α residues $^{289}$LMC$^{291}$. ASPP2$^{R921}$ is engaged in a salt bridge with PP1α$^{D242}$. **c** Close-up view of ASPP2 Ankyrin1 domain interaction with PP1. **d** The PP1α PPII motif binds to ASSP2 SH3 domain. The three core residues PP1α$^{P318}$, PP1α$^{P321}$ and PP1α$^{R323}$ of the PxxPxR motif are engaged in an extensive network of hydrophobic and electrostatic interactions. The side chain of PP1α$^{R317}$ forms an additional water-bridged hydrogen bond with the backbone of ASPP2$^{E1069}$. We also observed three hydrogen bonds between the PP1α mainchain and the ASPP2 SH3 domain: two between the carbonyls of PP1α$^{P321}$/ PP1α$^{P322}$ and the side chain of ASPP2$^{W1097}$, and one between PP1α$^{I319}$ carbonyl and ASPP2$^{N1112}$ side chain. **e** Schematic of PP1 C-tail interaction with ASPP2 SH3 domain. Red shading represents the negatively charged ASPP2 specificity pocket. The last three PP1α lysine residues are shadowed in blue

**The extreme PP1α C-tail interacts directly with ASPP2.** Although our ITC data support the importance of the PP1α C-tail lysines in ASPP2 binding, these residues were not detectable in our crystal structure. To identify the PP1α C-tail residues implicated in ASPP2 binding, we isotopically labelled ($^{15}$N) the PP1α C-tail for NMR spectroscopy. First, we acquired the 2D [$^{1}$H,$^{15}$N] HSQC spectrum in its unbound form. The spectrum displayed all the hallmarks for a peptide including no chemical shift dispersion in the $^{1}$H dimension due to lack of secondary structure elements (Fig. 7a). Next, we performed the sequence-specific backbone assignment of the PP1α C-tail (97% complete) using a $^{13}$C, $^{15}$N-labelled peptide. The secondary chemical shift (CSI) plot confirmed the lack of preformed secondary structure elements in the free PP1α C-tail (Supplementary Fig. 8).

We then titrated unlabelled ASPP2 into $^{15}$N-labelled PP1α C-tail in order to observe either chemical shift perturbations or peak intensity changes that are indicative of binding (Fig. 7b). Indeed, PP1α C-tail cross peaks either shifted or had dramatically reduced peak intensities (e.g. PP1α$^{310-325}$) upon ASPP2 titration. This indicates that these residues are the core binding residues, correlating well with those observed in the electron density of the PP1α:ASPP2 complex (PP1α$^{317-323}$). Interestingly, the PP1α C-tail residues that flank the core binding domain also showed chemical shift perturbations, indicating that they also contribute to binding. Thus, to comprehensively identify all interacting residues, we recorded a $^{15}$N[$^{1}$H] heteronuclear NOE and a $^{15}$N transverse relaxation ($R_2$) experiment of the PP1α C-tail in the free and ASPP2 bound states. These experiments showed that all PP1α C-tail residues, including the three C-terminal lysines, have reduced flexibility in the presence of ASPP2 (Fig. 7c, d). Thus, these lysines clearly contribute to the overall binding between these two proteins, as also confirmed by ITC (Fig. 6c), and do so via a fuzzy charge–charge interaction (lysines can flexibly move about the negatively charged pocket provided by the n-Src loop and thus contribute to binding without the need of locking into a single conformation, which would be visible in the crystal structure).

**Function of the ASPP2 SH3 domain/PP1α C-tail interaction.** To test the role of the SH3 acidic n-Src loop in the context of the full ASPP/PP1 complex, we used an AP-MS approach. We stably expressed Strep-HA tagged ASPP2 or ASPP2$^{KVK}$ (ASPP2$^{D1091K/}$$^{E1092V/E1094K}$) in HEK 293T cells using the Flp-In system (see Methods). After pulldown of ASPP2 or ASPP2$^{KVK}$ from cell lysates, we used Mass Spectrometry to quantify associated peptides (Fig. 8a). In agreement with the biophysical data, ASPP2 was associated with all the PP1 isoforms, and the ASPP2$^{KVK}$ mutation significantly decreased the amount of PP1α binding to ASPP2, while binding to PP1β and PP1γ1 was not affected (Fig. 8a). Indeed, PP1α was the only interactor of ASPP2 reduced upon mutation of the n-Src loop (Supplementary Fig. 9).

To address PP1 isoform specificity in a cellular context, we stably expressed Strep-HA tagged PP1α, PP1α$^{ΔC}$ (a truncation

lacking amino acids 301–330, including the PPII and C-tail lysines), PP1β and PP1γ1 in HEK293T and performed AP-MS experiments (Fig. 8b). As observed using ITC and BLI, ASPP2 showed a marked preference for PP1α, compared to PP1β and PP1γ1, while this selectivity was not observed for ASPP1 (Fig. 8b). iASPP does not discriminate between PP1α and PP1β, but shows low association to PP1γ1 (Fig. 8b). As expected, deletion of the C-tail of PP1α (comprising both the PPII and C-tail lysines) severely impaired binding between PP1α and all ASPP isoforms (Fig. 8b, c). In addition, binding of the other members of the ASPP/PP1 complex, RASSF7/8 and CCDC85 are severely reduced by the PP1α$^{ΔC}$ truncation, clearly indicating the importance of the PP1 C-tail in the assembly of the ASPP/PP1 tetrameric complex (Fig. 8c).

As the ASPP2$^{KVK}$ mutation specifically disrupted PP1α recruitment to the ASPP2/PP1 complex (Fig. 8a), we tested the effect of this mutation on the dephosphorylation of TAZ, a known substrate of the complex[36]. As previously reported, overexpression of wild type ASPP2 in HEK293T cells promotes TAZ dephosphorylation on Serine 89, which is prevented by mutation of the RVxF SLiM (Fig. 8d, e and ref. [36]). Interestingly, ASPP2$^{KVK}$ was also compromised in its ability to dephosphorylate TAZ (Fig. 8d and e), suggesting that the interaction between the ASPP2 SH3 domain specificity pocket and PP1α is required for ASPP2/PP1 substrate dephosphorylation.

**The SH3 domain is required for ASPP in vivo function.** To test the in vivo effect of disrupting the interaction between the ASPP SH3 and PP1, we generated ASPP rescue constructs with mutations in the RVxF motif (ASPP$^{FA}$), the SH3 domain (ASPP$^{WK}$) and a combination of mutations in both domains (ASPP$^{FA-WK}$). Expression of wild type ASPP in ASPP mutant flies fully rescues the increased number of IOCs and bristle misplacement (Supplementary Fig. 10a–c, g, h). The RVxF mutant, (ASPP$^{FA}$) did not fully rescue the IOC and bristle phenotypes, and mutation of both the RVxF and SH3 (ASPP$^{FA-WK}$) enhanced this effect (Supplementary Fig. 10a–d, f–h). In contrast, the SH3 specificity pocket mutant ASPP$^{KVK}$ (ASPP$^{D981K/D982V/E984K}$) was able to rescue the increase in IOC number phenotype, though bristle misplacements were still apparent (Supplementary Fig. 10e, g and h). The relatively mild effect of the ASPP$^{KVK}$ mutation was surprising, since the equivalent mutation in ASPP2 strongly reduced affinity towards PP1α (Figs. 6d and 8a).

Since *Drosophila* Ccdc85 can associate with PP1 in cell culture (Fig. 1d), we hypothesized that it might provide an extra contact point between the ASPP complex and PP1, thereby partially rescuing the effect of ASPP/PP1-binding surface mutations. Consistent with this hypothesis, mutation of *ASPP* and *ccdc85* together dramatically increased the IOCs number and the bristle misplacement phenotypes (Fig. 9a–c and i–j). These defects could be rescued back to *ccdc85* single mutant levels by the expression of the wild type form of ASPP in an *ASPP*, *ccdc85* mutant background (Fig. 9d and i–j). In contrast, all the *ASPP* mutant forms (ASPP$^{FA}$, ASPP$^{WK}$, ASPP$^{KVK}$, ASPP$^{FA-WK}$) failed to rescue

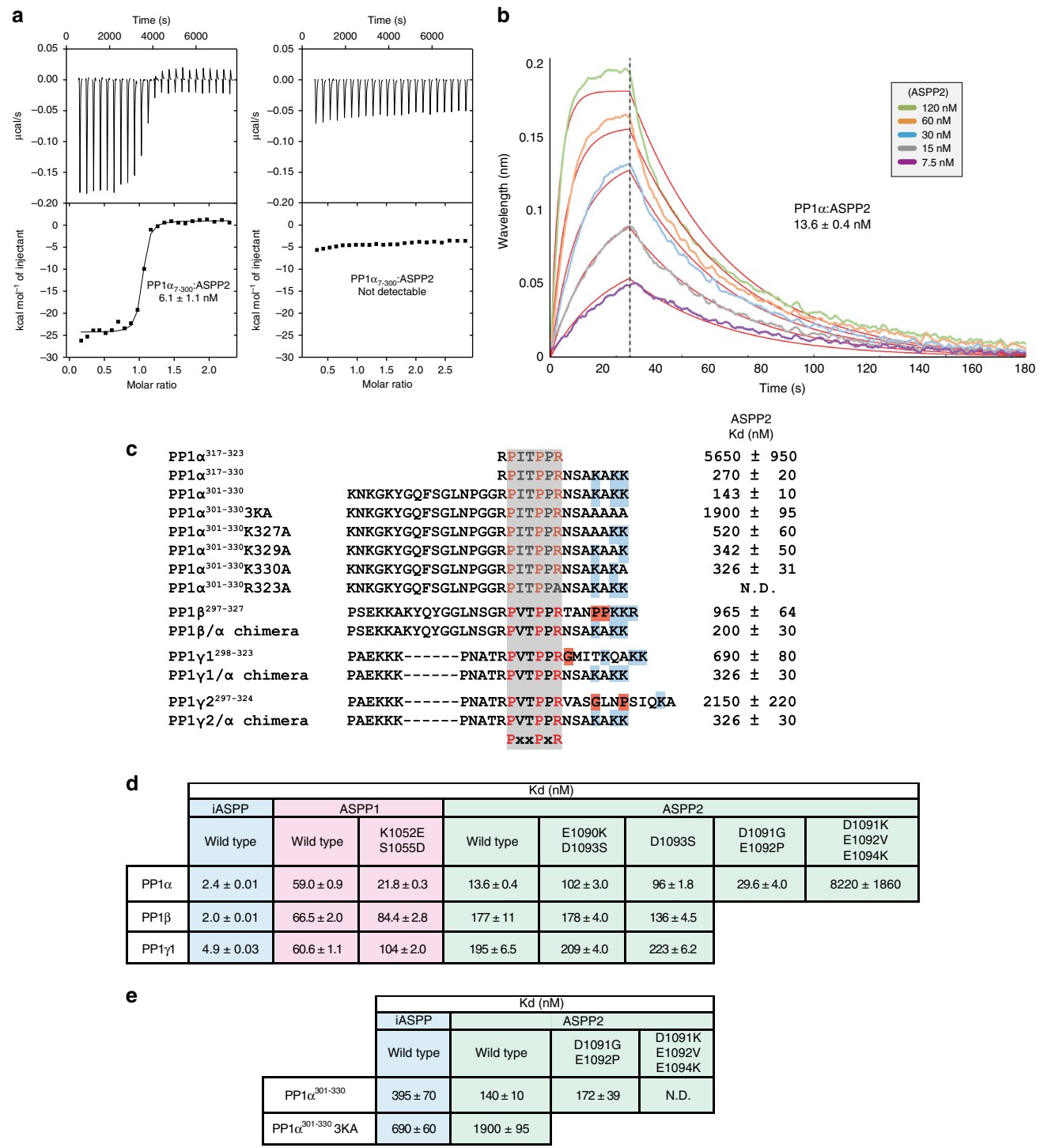

**Fig. 6** Affinity measurement for PP1α:ASPP2 complex. **a** ITC measurement of ASPP2$^{920-1120}$:PP1α$^{7-330}$ and ASPP2$^{920-1120}$:PP1α$^{7-300}$. **b** BLI measurement of ASPP2$^{920-1020}$ binding to immobilized PP1α$^{7-330}$. **c** Affinity measurement of various human PP1 C-tail peptides by ITC. The PPII motif is highlighted in grey, glycine and proline residues in the C-terminal region are highlighted in red and C-terminal basic residues in blue. **d** Affinity measurement of various PP1: ASPP complexes by BLI. The following proteins were used for these assays: ASPP2$^{920-1120}$, ASPP1$^{882-1090}$, iASPP$^{621-828}$, PP1α$^{7-330}$, PP1β$^{6-327}$, PP1γ$^{7-323}$. **e** ITC measurement of various PP1α$^{301-330}$:ASPP complexes

the IOC number and bristle misplacement phenotypes (Fig. 9e–h and i–j). Thus, consistent with our binding data, both the RVxF and the SH3 domain are important for ASPP function in vivo.

**The ASPP complex promotes PP1β9C and PP1α96A AJ localisation.** *Drosophila* ASPP preferentially binds to two PP1

isoforms, PP1α96A and PP1β9C (Fig. 1b). Analysis of GFP tagged forms of these two isoforms expressed under their endogenous promoter revealed an adherens junction pool colocalised with E-cadherin in retinas at 26 h APF (Supplementary Fig. 10k–n), where we have previously shown ASPP and RASSF8 to be localised[31]. In contrast, GFP tagged PP1α87B, which has low affinity

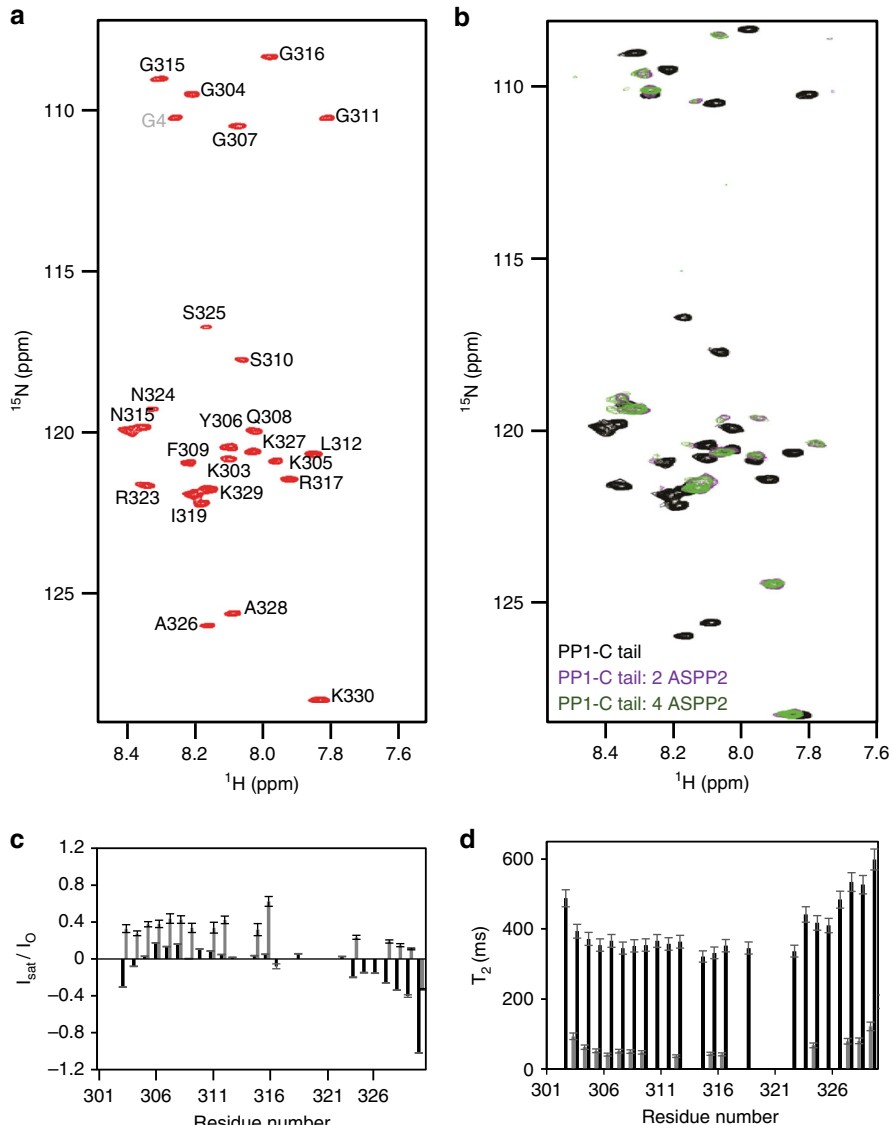

**Fig. 7** A fuzzy charge–charge interaction enhances PP1α C-tail binding to ASPP2. **a** Fully annotated 2D [$^1$H,$^{15}$N] HSQC of PP1$_{301-330}$. **b** 2D [$^1$H,$^{15}$N] HSQC of PP1$^{301-330}$ (black) in presence of a 2 (violet) or 4 (green) molar excess of ASPP2. **c** Comparative analysis of $^{15}$N[$^1$H] NOE of free PP1α C-tail (black bars) and PP1α C-tail: ASPP2 (grey bars); missing peaks are due to overlap, making quantitative intensity analysis impossible. Error bars represent standard deviations. **d** Comparative analysis of $^{15}$N transverse relaxation (R$_2$) data of free PP1α C-tail (black bars) and PP1α C-tail: ASPP2 (grey bars). Error bars represent standard deviations

for ASPP (Fig. 1b), localises in the cytoplasm (Supplementary Fig. 10i–j). To test whether the junctional pool of PP1α96A and PP1β9C is dependent on the ASPP complex, we examined their localisation in *ASPP, ccdc85* double mutant animals (Fig. 9k–t and Supplementary Fig. 10o–u). Interestingly, junctional localisation of PP1α96A and PP1β9C was disrupted in *ASPP, ccdc85* double mutant (Fig. 9k–m, q–t and Supplementary Fig. 10o–u), while *ASPP* single mutants showed unaltered PP1β9C localisation (Fig. 9n–p and t). Thus, the ASPP complex is required to recruit a junctional pool of PP1β9C and PP1α96A.

## Discussion
How highly specific phosphatase complexes are assembled to induce precise spatially and temporally controlled dephosphorylation of kinase substrates remains a key open question with important therapeutic ramifications[53,54]. Here, we show that ASPP proteins form a PP1-containing complex with Ccdc85 and

members of the N-terminal RASSF family (Figs. 1, 8 and S1). Depletion of these proteins in *Drosophila* leads to similar phenotypes and mutations in the ASPP PP1-binding motifs compromise its in vivo functions (Figs. 2, 3, 9, Supplementary Fig. 2, Supplementary Fig. 3, Supplementary Fig. 10). Our structural and biophysical data indicate that human ASPP2 can interact with PP1 catalytic subunits both through its RVxF SLiM and its SH3 domain, which binds an extended PPII motif in the PP1 C-tail (Figs. 5–7). Interestingly, at least in *Drosophila*, Ccdc85 appears to provide an additional PP1-binding surface to the ASPP/PP1 complex. Indeed, the phenotypes elicited by mutations in the ASPP RVxF motif or SH3 domains are markedly enhanced by loss of *ccdc85* (Fig. 9). Our structure indicates that ASPP proteins do not select appropriate substrates for dephosphorylation by restricting access to the PP1 catalytic cleft (Fig. 5), as reported for some PIPs such as Spinophilin[14]. Instead, in vivo analysis suggests that the ASPP complex is required for the recruitment of a

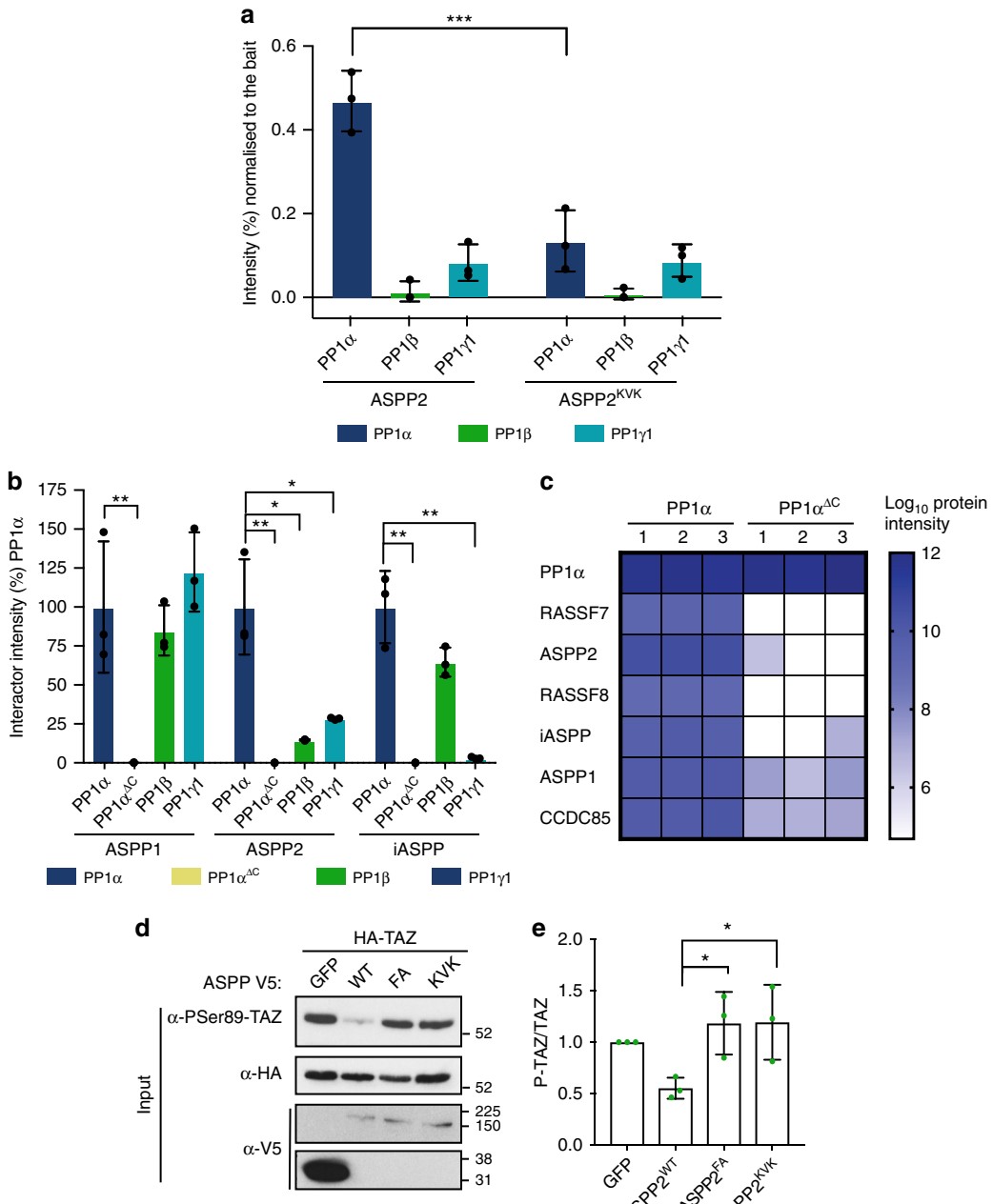

**Fig. 8** Function of the ASPP2 SH3 domain/PP1α C-tail interaction. **a** Quantitative AP-MS from HEK293T cells expressing Strep-HA tagged ASPP2 or ASPP2[KVK]. Chart shows protein abundance normalised to the bait (%) relative to PP1α, PP1β, PP1γ1. Protein abundance is measured on the basis of the average intensity of the three most intense and unique peptide precursors. Error bars indicate the standard deviation. A two-way ANOVA test was done to determine if the differences among means were significantly different from each other. Three pairwise comparisons were carried out (ASPP2 PP1α vs ASPP2KVK PP1α, ASPP2 PP1β vs ASPP2[KVK] PP1β and ASPP2 PP1γ1 vs ASPP2[KVK] PP1γ1) and p-values were adjusted using Bonferroni correction. Significant differences are marked. *** indicates $p < 0.0001$. **b** Quantitative AP-MS from HEK 293T cells expressing Strep-SH tagged PP1α, PP1α[ΔC] (PP1α[1-300]), PP1β and PP1γ1. Protein abundance of ASPP1, ASPP2 and iASPP is measured on the basis of the three most intense and unique peptide precursors. Each intensity value is normalised to the respective bait and to PP1α. Error bars indicate standard deviation. A two-way ANOVA was done to determine if differences among means were significantly different from each other. Three pairwise comparison among each group were carried out (PP1α vs PP1αΔC, PP1α vs PP1β and PP1α vs PP1γ1) and p-values were adjusted using a Bonferroni correction. Significant differences are marked. * indicates $p < 0.05$, ** indicates $p < 0.005$. **c** Comparison of PP1α and PP1α[ΔC] protein interactions. Heatmap of ASPP2-PP1 complex proteins intensity after purification with PP1α or PP1α[ΔC]. **d** Western blot of transfected HEK293T cell lysates probed with the indicated antibodies. **e** Quantification of the ratio between P-TAZ[S89] and total HA-TAZ protein levels normalised to control phosphorylation. Error bars represent standard deviation. A one-way ANOVA test was carried out to determine if the differences among means were significantly different from each other. Two pairwise comparisons were carried out (ASPP2[wt] vs ASPP[FA] and ASPP[wt] vs ASPP[KVK]) and p-values were adjusted using a Bonferroni correction. Significant differences are marked. * indicates $p < 0.05$ ($n = 3$ independent experiments)

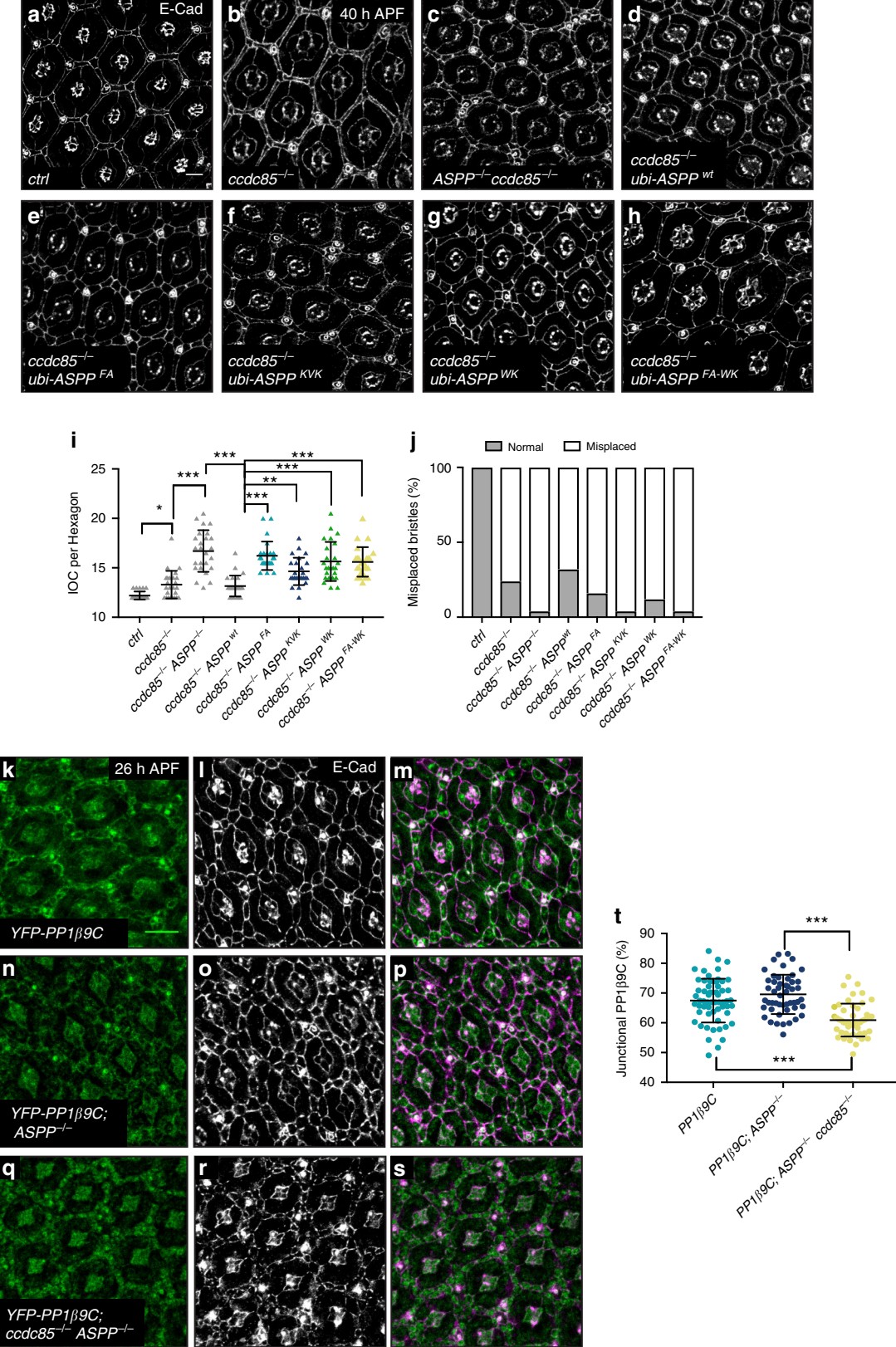

junctional pool of PP1α96A and PP1β9C (Fig. 9, Supplementary Fig. 10).

The structure of the ASPP2:PP1α complex represents observation of the PP1 C-tail interacting with a PIP and providing isoform specificity. Since most PP1-binding motifs are SLiMs present within intrinsically disordered regions, this constitutes a rare occurrence of a folded protein domain involved in PP1 recruitment[2,12]. The ASPP2 SH3 domain accommodates the highly conserved PP1α PPII motif while its acidic specificity pocket enhances affinity and specificity for PP1α C-tail basic

**Fig. 9** The ASPP complex promotes PP1β9C and PP1α96A AJ localization. **a–h** Confocal X–Y sections of pupal retinas at 40 h APF stained with anti-E-cad antibodies to mark cell outlines. Scale bar = 10 μm. **a, b** ccdc85 (**b**) mutant flies have defects in eye development compared to wild-type flies (**a**) (13.30 ± 1.40 vs 12.2 ± 0.4). **c** Mutation of ASPP in ccdc85 mutant flies worsens the phenotype (16.7 ± 2.10). **d–g** Expression of ASPP under the ubiquitin 63E (ubi) promoter in ASPP, ccdc85 mutants rescues the phenotype (**d**), but overexpression of $ASPP^{FA}$ (**e**), $ASPP^{KVK}$ (**f**), $ASPP^{WK}$ (**g**) or $ASPP^{FA-WK}$ (**h**) do not (13.16 ± 1.058 vs 16.22 ± 1.44, 14.64 ± 1.38, 15.64 ± 1.98 and 15.60 ± 1.49) (see Supplementary table 1 for full genotypes). **i** Quantification of inter-ommatidial cells (IOCs) in retinas of the indicated genotypes at 40 h after puparium formation (APF). Error bars represent the standard deviation. A one-way ANOVA test was carried out to determine if the differences among means were significantly different from each other. p-values were adjusted using a Bonferroni correction. Significant differences are marked. *** indicates $p < 0.0001$, ** indicates $p < 0.01$ and * indicates $p < 0.05$ ($n = 25$ hexagons from four retinas). **j** Quantification of bristle misplacement. ($n = 25$ from four retinas). **k–s** Confocal X–Y sections of pupal retinas at 26 h APF showing endogenous YFP-tagged PP1β9C and stained with E-Cad. Scale bar = 10 μm. **k–m** PP1β9C localises at the cell–cell junctions (67.31% ± 7.45). **n–p** ASPP loss does not change the localisation of PP1-β9C (69.23% ± 6.29). **q–s** Loss of ccdc85 in an ASPP mutant background mislocalises PP1β9C from the cell–cell junctions (60.92% ± 5.51). **t** Quantification of percentage of junctional PP1β9C of the indicated genotypes at 26 h APF. Error bars represent the standard deviation. For quantifications of PP1β9C intensity, two regions of interest were drawn surrounding the junctions of a cell ($ROI_{TOTAL}$ and $ROI_{CYTOPLASM}$). Junctional fraction was quantified by ($ROI_{TOTAL} - ROI_{CYTOPLASM}$). Values were normalised to 100% for PP1β9C. One-way ANOVA test was carried out to determine if the differences among means were significantly different from each other. p-values were adjusted using a Bonferroni correction. Significant differences are marked. *** indicates $p < 0.0001$ ($n = 50$ cells from four retinas)

residues $PP1\alpha^{K327}$, $PP1\alpha^{K329}$ and $PP1\alpha^{K330}$ (Figs. 5e, 6c, d), through fuzzy electrostatic interactions (Fig. 7). The variability observed in this region for the other PP1 isoforms results in reduced affinity. The first lysine ($PP1\alpha^{K327}$) appears the most critical for ASPP2 binding (Fig. 6c), suggesting that the spacing between the PPII and C-terminal lysines is important for ASPP2 selectivity towards PP1α, which has the shortest spacer of all PP1 isoforms (Fig. 4a). In addition, the presence of residues such as glycines or prolines in the PP1β and PP1γ spacers, which likely alter or constrain C-tail geometry also reduce affinity towards ASPP2, as suggested by chimera experiments (Fig. 6c).

Similar differences in affinity are sufficient to achieve PP1 isoform selectivity in vivo, as illustrated for RepoMan/Ki-67, which selectively bind to PP1γ based on a single amino acid change in the PP1 catalytic domain[11]. Indeed, our AP-MS data show that mutating the ASPP2 SH3 domain specificity pocket ($ASPP2^{KVK}$) dramatically reduces binding to PP1α but not other isoforms (Fig. 8a). Furthermore, ASPP2 shows a marked preference for PP1α in HEK293T cells, while ASPP1 does not (Fig. 8b). It is interesting to note, however, that ASPP1 has a phosphorylatable residue (S1055) in its n-Src loop (Supplementary Fig. 7A), therefore ASPP1 isoform selectivity might be tuned by phosphorylation. iASPP shows modest selectivity towards PP1α and PP1β versus PP1γ1 in vitro (Fig. 6d), and this preference is even more marked in HEK293T cells, where PP1γ1 association is comparatively weak (Fig. 8b). This increased selectivity in a cellular context could be due to the presence of other iASPP binding partners in HEK293T cells or the fact that the BLI experiments (Fig. 6d) were performed on an N-terminally truncated form of iASPP ($iASPP^{621-828}$).

The composition of the ASPP2 SH3 domain specificity pocket allows discrimination between different PP1 isoforms (Fig. 6c, d), providing isoform selectivity through the PP1 PPII motif. Interestingly, in Drosophila, isoform selectivity though the C-tail operates in rather dramatic fashion, since two PP1 isoforms (PP1α87B and PP1α13C) lack the PPII motif altogether (Fig. 4a). Given the diversity in specificity pockets among SH3 domains[55], we propose that different SH3s may dictate different PP1 isoform preferences depending on the number and exact positioning of acidic or hydrophobic residues in their RT and n-Src loops forming the specificity pocket. Thus, it is possible that some PIPs and/or substrates of PP1 holoenzymes use an SH3 domain to associate with PP1, providing much scope for the expansion of the PP1 regulatory and substrate pool.

The structures of ASPP2 in complex with the p53 and p73 tumour suppressors reveal a direct interaction between the ASPP2 SH3 domain and the p53/p73 DNA binding domains centred on the cancer hotspot residues $p53^{R248}$ and $p73^{R268}$ [56,57]. Those arginines are both engaged in a network of electrostatic interactions with the same three acidic residues that coordinate $PP1\alpha^{R323}$ in our structure (Supplementary Fig. 7b, c). This provides an explanation for the mutually exclusive binding of PP1 and p53 to ASPP2 reported previously[25,26].

In conclusion, we have identified a mode of PP1/PIP association where the ASPP2 SH3 domains engages the PP1 C-tail via an extended PPII and is able to discriminate between PP1 isoforms based on the sequence and length of their C-tails. ASPP1/2 have been proposed to promote the activity of the pro-tumour YAP/TAZ transcription co-activators by dephosphorylating an inhibitory residue phosphorylated by the Hippo core kinase cascade[35,36]. Given the current interest in inhibiting YAP/TAZ activity for cancer therapy[58], inhibitors of the ASPP2/PP1 complex, which would increase YAP/TAZ inhibitory phosphorylation, might prove an interesting therapeutic avenue. Our work indicates that the ASPP2 SH3 domain constitutes an attractive target for the specific inactivation of the ASPP/PP1 holoenzyme. Indeed, mutation of the ASPP2 SH3 domain specificity pocket ($ASPP2^{KVK}$) compromises the dephosphorylation of TAZ at its inhibitory site, serine 89 (Fig. 8). The ability to target specific SLiM interaction pockets in Ser/Thr phosphatases is supported by the discovery that the widely used immunosuppressants cyclopsorin A and FK506 bind calcineurin in the LxVP SLiM binding pocket[59]. Thus, a better understanding of the PP1 regulatory code will impact future drug discovery efforts[54,60].

## Methods

**Cell culture.** Drosophila S2 cells (Drosophila Genome Resource Center; RRID: CVCL_Z232) were maintained at 25 °C in Schneider's Drosophila Medium (Gibco) supplemented with 10% fetal bovine serum (Sigma-Aldrich) and 100 unit/ml penicillin and 100 mg/ml streptomycin (Gibco) in 75 cm² cell culture flasks (Corning). For transfections, 30 min prior to transfection, $3 \times 10^6$ S2 cells were seeded per well of a 6-well plate (Corning) in 2 ml of serum-containing Schneider's Drosophila Medium (Gibco). Two hundred to four hundred nanograms of DNA per plasmid were transfected per well using Effectene Transfection Reagent (Qiagen) according to the manufacturer's protocol. Cells were lysed 48 h after transient transfection. Human embryonic kidney cells (HEK293T) (Crick Cell Services) were cultured in 5% $CO_2$ atmosphere and 37 °C in DMEM (Dulbecco's modified Eagle's medium, SIGMA) supplemented with 10% FBS (fetal bovine serum, Gibco) and penicillin/streptomycin (100 μg/ml, Gibco). Cells were transfected with Lipofectamine ® 2000 (Thermo Fisher) according to the manufacturer instructions. Cells were lysed 48 h after transfection. Cell lines were profiled by STR and tested negative for mycoplasma by the Cell Services Technology Platform at the Francis Crick Institute.

**Co-IPs and western blotting**. S2 and HEK293T cells were lysed with HEPES lysis buffer (150 mM NaCl, 50 mM HEPES pH 7.5, 0.5% (v/v) Triton X-100 supplemented with protease inhibitor cocktail with EDTA (Roche) and the soluble fraction was obtained by centrifugation (16,000 × g, 20 min). For co-IP experiments cleared lysates were added to 20 μL of ANTI-FLAG M2 Affinity Gel (Sigma-Aldrich) or 15 μL of GFP-Trap agarose (ChromoTek) in 100 μL HEPES lysis buffer. The lysate and the beads were incubated for 1 h at 4 °C and washed four times with HEPES lysis buffer. Input and co-IP samples were analysed by SDS-PAGE and western blot. Primary antibodies were: mouse anti-FLAG (M2), 1/5000 (Sigma-Aldrich F1804); mouse anti-GFP (3E1), 1/1000 (Cancer Research UK); rat anti-HA (3F10), 1/2000 (Roche); mouse anti-Myc (9E10), 1/5000 (Santa Cruz sc:40); mouse anti-Tubulin (E7), 1/5000 (Developmental Studies Hybridoma Bank); mouse anti-V5 (R960-25), 1/5000 (Thermo Fisher), rabbit anti-P-TAZ$^{Ser89}$ 1/1000 (E1X9C) (Cell Signalling Technologies 59971), rabbit anti-HA 1/1000 (C29F4) (Cell Signalling Technologies 3724). Secondary antibodies were from GE Healthcare (1:5000).

**Molecular biology**. Fly genomic DNA was isolated using the DNeasy Blood & Tissue Kit (Qiagen). For RT-PCR, total RNA was isolated using the RNeasy Mini Kit (Qiagen) and treated with DNase (Promega) for 37 °C for 30 min before adding DNase Stop Solution (Promega). cDNA was synthesised from 1 μg of isolated RNA with oligo(dT) primers using the First Strand cDNA Synthesis Kit for RT-PCR (Roche) according to the manufacturer's instructions.

All _Drosophila_ S2 and human cell expression vectors were obtained using Gateway cloning (Invitrogen). Entry vectors were generated by PCR into the Gateway pDONR/Zeo Vector (Invitrogen) or pENTR™-TOPO (Invitrogen) using Gateway BP clonase II Enzyme mix (Invitrogen) or TOPO cloning (Invitrogen) according to the manufacturer's protocol. Site-directed mutagenesis was performed using PfuTurbo DNA Polymerase (Stratagene) or were previously described[31]. Expression vectors were obtained using Gateway LR clonase II Enzyme mix (Invitrogen) according to manufacturer's protocol into _Drosophila_ Gateway Vector Collection plasmids (_Drosophila_ Genomics Resource Center), pDest-cDNA-X-V5, pDEST-3xFLAG-CMV-7.1 or pCDNA5/FRT/TO/SH/GW[61]. To generate the ASPP rescue constructs, full-length ASPP cDNAs were cloned using the Gateway system into pKC26w-pUbiq[32]. The resultant plasmids were injected into _PBacy[+]-attP-9AVK00018_ (attP integration at 53B2) by Bestgene Inc. UAS lines for _ASPP_ and _ccdc85_ were generated by cloning the cDNAs into pUASg.attB or pUASg-HA. attB[62] and injected into either _PBacy[+]-attP-3BVK00002_ (ccdc85) or _PBacy [+]-attP-9AVK00018_ (ASPP) by Bestgene Inc. To generate ubi-GFP-PP1α87B, full-length PP1α87B was cloned using the Gateway system into pKC26w-pUbiq. The resultant plasmid was injected into VIE-217 by Bestgene Inc. Human ASPP2$^{920-1120}$ was amplified from ASPP2 cDNA (kind gift from Xin Lu), cloned into TOPO TA vector (Invitrogen) according to the manufacturer instruction and was later cloned into PGEX6p2 vector (GE Healthcare) using BamHI and NotI restriction enzymes (NEB Biolabs). Mutagenesis of the plasmids was performed with the indicated primers using PWO Master Mix (Roche) followed by DpnI incubation for one hour. The same procedure was followed to clone ASPP1$^{882-1090}$. PP1α (26566), PP1β (51769), PP1 γ1 (51770) and TAZ (32839) plasmids were obtained from Addgene. pET47b-iASPP$^{621-828}$ was a kind gift from Mark Glover. See Supplementary table 2 for primers used in this study.

**Fly genetics**. To generate the _ccdc85_ mutant, a transposon (P-element P{XP}d06579), inserted in the _ccdc85_ 5′ UTR _ccdc85_ (CG17265) locus was imprecisely excised by crossing to a transposase source and screening for loss of the _w⁺_ eye pigmentation marker in the P-element. These selected flies were then genotyped for deletions within the _ccdc85_ gene using PCR (Fig. S2a, b). We isolated one excision, _C1.1_, which removed the entire first exon of the _ccdc85_, and therefore named this mutant stock _ccdc85$^{C1.1}$_. _ccdc85$^{C1.1}$_ animals were viable but displayed a rough eye phenotype similar to that of _ASPP_ and _RASSF8_ mutants, either as homozygotes or in _trans_ to a deficiency (_Df(2L)Exel7014_) or the original _P{XP}d06579_ transposon insertion. RT-PCR experiments revealed that _ccdc85_ transcripts are undetectable in _ccdc85$^{C1.1}$_ animals (Supplementary Fig. 2c). Together with the fact that the rough eye phenotype of _ccdc85$^{C1.1}$_ homozygotes is identical to that of _ccdc85$^{C1.1}$/Df(2L)Exel7014_ animals, this suggests that _ccdc85$^{C1.1}$_ represents a null allele of _ccdc85_. _ccdc85$^{C1.1}$_ also removes part of the 3′ UTR of the neighbouring gene, _CG3558_, and transcript levels for _CG3558_ were slightly reduced in mutant animals, suggesting that _ccdc85$^{C1.1}$_ might also be a _CG3558_ hypomorph (Supplementary Fig. 2c). FLP/FRT clones were generated using the Flp/FRT system using heat shock-driven Flipase (_hsFLP_). Heat shock were performed at 37 °C for 1 h at 48 h after embryo deposition.

**Fly stocks**. The following fly lines were described before _ASPP$^{2.93}$_, _ASPP[1]_, _w; FRT 42D ASPP$^d$_, _ASPP[8]_, _FRT 82B, Csk1jd8[31,63]_ and _Df(2L)Exel7014[64]_, _GFP-PP196A_ (FlyFos021765(pRedFlp-Hgr)(Pp1alpha-96A15346::2XTY1-SGFP-V5-preTEV-BLRP-3XFLAG)dFRT (VDRC ID 318084))[65] and _PBac{681.P.FSVS} flwCPTI001360[66,67]_. See all the fly genotypes in Supplementary Table 1.

**Immunofluorescence**. Third instar larval wing imaginal discs or pupal retinas were dissected in PBS and fixed with 4% formaldehyde in PBS at room temperature for 20 min. After fixation, the tissue was washed twice with PBT, permeabilised

with PBS containing 0.3% (v/v) Triton X-100 (PBT) for 30 min at room temperature, blocked with PBT containing 10% normal goat serum (NGS) for 30 min at room temperature and incubated at 4 °C overnight in primary antibody. Following five washes in PBT, the tissue was incubated for 1 h at room temperature in secondary antibody, then washed and mounted in Vectashiled with DAPI (Vector) on glass slides. Primary antibodies used: rat anti-E-cadherin (DCAD2), 1/100 (Developmental Studies Hybridoma Bank), rat anti-ASPP, 1/500[31], mouse anti-β-Galactosidase, 1/500 (Promega). Images were acquired on a Zeiss LSM 710 confocal laser-scanning microscope or a SP5 laser scanning confocal (Leica). Images were processed with ImageJ.

**Cell line generation and AP-MS analysis**. HEK Flp-In293 T-Rex cells (Thermo Fisher) containing a single genomic FRT site and stably expressing the tet repressor were cultured in DMEM medium (4.5 g/l glucose, 2 mM L-glutamine; Thermo Fisher) following manufacturer's specifications. For cell line generation, Flp-In HEK293 cells were co-transfected with the corresponding expression plasmids (ASPP1, ASPP2, ASPP2$^{KVK}$, PP1α, PP1α$^{ΔC}$, PP1β, PP1γ) and the pOG44 vector (Thermo Fisher) for co-expression of the Flp-recombinase using FuGENE 6 transfection reagent according to manufacturer's specifications (Promega). Two days after transfection, cells were selected in hygromycin-containing medium (100 μg/ml). Stable isogenic cell pools were grown in four 14 cm Nunclon dishes, induced with 1.3 mg/ml doxycline for 24 h for the expression of SH-tagged bait proteins, harvested with PBS and frozen in liquid nitrogen. Each cell line was processed for three biological replicates. The frozen cell pellets were resuspended in 3 ml HNN lysis buffer [50 mM HEPES pH 7.5, 150 mM NaCl, 50 mM NaF, 0.5% Igepal CA-630, 200 μM Na$_3$VO$_4$, 1 mM PMSF, 20 μg/ml Avidin and 1X Protease Inhibitor mix (Sigma-Aldrich)] and incubated on ice for 10 min. Cleared lysates were incubated with 100 μl slurry Strep-Tactin sepharose beads (IBA GmbH) in spin column (Bio-Rad). The beads were washed with 2 ml HNN lysis buffer and 3 ml of HNN buffer (50 mM HEPES pH 7.5, 150 mM NaCl, 50 mM NaF). Bound proteins were eluted with 600 μl 0.5 mM biotin in HNN buffer and processed following the FASP protocol[68]. Briefly, the eluate was loaded on a 10 kDa molecular weight cut-off spin column (Vivaspin 500, Sartorious) and centrifuged at 14,000 × g for 30 min until dryness. Samples were denatured, reduced (8 M Urea and 5 mM TCEP in 50 mM ammonium bicarbonate, 30 min) and alkylated (10 mM iodoacetamide, 30 min). Each sample was subsequently washed three times by flushing the column with 25 mM ammonium bicarbonate and proteolyzed with 0.5 μg of Trypsin (Promega, sequencing grade) for 16 h at 37 °C. Proteolysis was quenched by 0.1% TFA and peptides were purified with a C18 microspin column (Nest Group), dried using a speed vacuum before being resuspended in 20 μl 0.1% formic acid and 2% acetonitrile. iRT peptides (Biognosys) were spiked to each sample before LC-MS/MS analysis for quality control. LC-MS/MS was performed on Orbitrap Fusion Tribrid mass spectrometer (Thermo Fisher) coupled with a Thermo easyLC-1000 liquid chromatography system (Thermo Fisher). Peptides were separated using reverse phase column (NanoEase C18 100 Å, 1.8 μm, 75 μm*250 mm) across 60 min linear gradient from 5 to 35% (buffer A: 0.1% (v/v) formic acid, 2% (v/v) acetonitrile; buffer B: 0.1% (v/v) formic acid, 98% (v/v) acetonitrile). The data acquisition mode (data-dependent acquisition) was set to perform a cycle of 3 s with high resolution MS scan in the Orbitrap (120,000 at 400 _m/z_) and MS/MS spectra in the Ion Trap. Charge states lower than two and higher than seven were rejected. The dynamic exclusion window was set to 25 s. Precursors with MS signal that exceeded a threshold of 5000 were fragmented (HCD, Collision Energy 30%). The ion accumulation time was set to 50 ms (MS) and 80 ms (MS/MS).

Acquired spectra were searched using the MaxQuant software package version 1.5.2.8 embedded with the Andromeda search engine[69] against human proteome reference dataset (http://www.uniprot.org/) extended with reverse decoy sequences. The search parameter were set to include only full tryptic peptides, carbamidomethyl as static peptide modification, oxidation (M) and phosphorylation (S,T,Y) as variable modification. The MS and MS/MS mass tolerance was set, respectively, to 20 ppm and 0.5 Da. False discovery rate of <1% was used at the protein level to infer the protein presence. The protein abundance was determined from the intensity of top two unique peptides for each protein. Raw data are provided in Tables 2 and 3.

**Image analysis**. The cell outlines of 40 h APF retinas were visualised using anti-E-cadherin antibodies. IOCs were counted within a hexagonal region of interest using ImageJ. Each vertex of the hexagon was placed in the centre of the six ommatidia surrounding one central ommatidium. Cells that were bisected by the lines of the hexagon were counted as half a cell. Bristle clusters were not counted as IOCs. Bristle displacement was defined as any hexagon containing either two bristle clusters directly next to each other or when two bristle clusters were not separated by tertiary pigment cells. Missing bristle clusters were not counted as bristle misplacement.

Adults fly wings were mounted in Euparal (Agar Scientific) and imaged with a LeicaDFC420C camera mounted on a Zeiss Axioplan2 microscope. ImageJ was used to trace the outline of wing blades.

Quantification of junctional YFP-PP1β9B and GFP-PP1α96A was performed using ImageJ. Statistical analysis were performed using the Prism Graphpad 7 software.

**Protein preparation**. An overnight pre-culture of PP1α[7-330], PP1α[7-300], PP1β[6-327], PP1γ1[17-323] was grown at 37 °C in LB medium supplemented with Kan, Cam and 1 mM MnCl₂. PP1 production was initiated by inoculating a 100 l fermenter with 400 ml of pre-culture. The cells were grown in LB medium supplemented with 1 mM MnCl₂ at 30 °C to an OD600 of ~0.5 when 2 g/l of arabinose was added to induce the expression of the GroEL/GroES chaperone. When OD600 reach ~1, the temperature was lowered to 10 °C and PP1:ASPP2 expression was induced with 0.1 mM IPTG for ~24 h. The cells were harvested by centrifugation, resuspended in fresh LB medium (again supplemented with 1 mM MnCl₂ and 200 μg/ml chloramphenicol to eliminate all ribosome activity) and agitated for 2 h at 10 °C. Harvested cells were resuspended in lysis buffer (50 mM Tris-HCl, pH 8.0, 5 mM imidazole, 700 mM NaCl, 1 mM MnCl₂, 1% TX-100, 0.5 mM TCEP, 0.5 mM AEBSF, 15 μg/ml benzamidine and complete EDTA-free protease inhibitor tablets) and lysed by French press. The supernatants is clarified by centrifugation and stored at −80 °C.

iASPP[608-828], ASPP1[882-1090], ASPP2[920-1120] sequence was inserted in a pGEX-6P2 plasmid (GE Healthcare) containing a cleavable GST tag. ASPP2[920-1120] expression was at 30 °C in *E. coli* Rosetta (DE3) pLysS (Merck). Bacteria were harvested by centrifugation and resuspended in lysis buffer (50 mM Tris-HCl, pH 8.0, 500 mM NaCl, 1% TX-100, 0.5 mM TCEP, 0.5 mM AEBSF and 15 μg/ml benzamidine). The fusion protein was batch-adsorbed onto a glutathione-Sepharose affinity matrix and ASPP2 recovered by cleavage with 3 C protease at 4 °C overnight in 50 mM Tris-HCl, pH 8.0, 250 mM NaCl, 0.5 mM TCEP. ASPP2[920-1120] was then further purified by size exclusion chromatography using a Superdex 200 column equilibrated and run in 50 mM Tris-HCl, pH 8.0 and 150 mM NaCl. Peptides for ITC measurements were synthesized by the LRI peptide synthesis core facility.

**Crystallization and data collection and refinement**. For PP1α:ASPP2 complex expression the plasmids encoding PP1α[7-330] and ASPP2[920-1120] were co-transformed with a pGRO7 plasmid encoding the chaperone GroEL/GroES into BL21 (DE3) *E. coli* cells. An overnight pre-culture of PP1:ASPP2 was grown at 37 °C in LB medium supplemented with Amp, Kan, Cam and 1 mM MnCl₂. PP1:ASPP2 production was initiated by inoculating a 100 l fermenter with 400 ml of pre-culture. The cells were grown in LB medium supplemented with 1 mM MnCl₂ at 30 °C to an OD600 of ~0.5 when 2 g/l of arabinose was added to induce the expression of the GroEL/GroES chaperone. When OD600 reach ~1, the temperature was lowered to 15 °C and PP1:ASPP2 expression was induced with 0.1 mM IPTG for ~18 h. The cells were harvested by centrifugation, resuspended in fresh LB medium (again supplemented with 1 mM MnCl₂ and 200 μg/ml chloramphenicol to eliminate all ribosome activity) and agitated for 2 h at 10 °C. Harvested cells were resuspended in lysis buffer (50 mM Tris-HCl, pH 8.0, 5 mM imidazole, 700 mM NaCl, 1 mM MnCl₂, 1% TX-100, 0.5 mM TCEP, 0.5 mM AEBSF, 15 μg/ml benzamidine and complete EDTA-free protease inhibitor tablets) and lysed by French press. The supernatants is clarified by centrifugation and stored at −80 °C. PP1:ASPP2 complex is then purified via Ni-NTA IMAC. PP1α:ASPP2 was eluted with 50 mM Tris pH 8.0, 200 mM Imidazole, 700 mM NaCl and 1 mM NiCl₂ at 4 °C. Then the complex was further purified via GST IMAC, the fusion protein was batch-adsorbed onto a glutathione-Sepharose affinity matrix and recovered by cleavage with 3 C protease at 4 °C overnight in 50 mM Tris-HCl, pH 8.0, 500 mM NaCl, 0.5 mM TCEP. The PP1α:ASPP2 complex was then further purified by size exclusion chromatography using a Superdex 200 column equilibrated and run in 150 mM CHES, pH 9.5, 500 mM NaCl and 0.5 mM TCEP. To grow crystals, the protein solution was concentrated to 30 mg/ml. The complex was crystallised at 20 °C using the sitting-drop vapour diffusion method. Sitting drops of 1 μl consisted of a 1:1 (vol:vol) mixture of protein and a well solution containing 0.1 M TRIS pH 8.5, 16% PEG 8000. Crystals appeared after 12 h and reached their maximum size after 36 h (1 mm × 0.5 mm × 0.02 mm). Crystals were cryoprotected in 100 mM TRIS pH 8.5, 150 mM CHES, pH 9.5, 150 mM NaCl, 20% PEG 8000, 0.5 mM TCEP and 20% glycerol. Crystals were flash-frozen in liquid nitrogen, and X-ray data sets were collected at 100 K at the I02 beamline of the Diamond Light Source Synchrotron (mx9826-26) (Oxford, UK).

Data collection and refinement statistics are summarized in Table 1. The data set was indexed, scaled and merged with xia2[70]. Molecular replacement was achieved by using the high resolution atomic coordinates of human PP1 from PDB 4M0V[43] in PHASER[71] Refinement was carried out by using Phenix[72]. Model building was carried out in COOT[73] Model validation used PROCHECK[74], and figures were prepared using the graphics programme PYMOL[75]. In the refined structure of PP1:ASPP2 100% of the residues were in favoured or allowed regions of the Ramachandran plot. The asymmetric unit contains two copies of the complex. The difference electron density map covering PP1 shows unambiguous density for residues 317–323 from PP1 C-tail for both copies of the complex. See supplementary materials for stereo view of a portion of the electronic density map.

**Isothermal titration calorimetry**. All the protein and peptide were dialysed overnight into 25 mM Tris pH 8.0, 200 mM NaCl, 0.5 mM TCEP at 4 °C. The peptide concentration was measured by measuring the peptide bond absorbance at 215 nm on a Denovix DS-11 FX + spectrophotometer. PP1 peptides (350–550 μM) were titrated into ASPP2[920-1120] (40 μM) using an ITC200 Microcalorimeter

(Microcal, Inc) at 20 °C. Titration curves were fitted using MicroCal Origin 7.0 software, assuming a single binding site mode.

His₆-tagged-PP1 (PP1α[7-330], PP1α[7-300]) used for ITC were purified as follows. PP1 was lysed and purified using Ni²⁺-affinity chromatography and SEC (pre-equilibrated ITC buffer, 20 mM Tris pH 8, 500 mM NaCl, 0.5 mM TCEP, 1 mM MnCl₂). ASPP2 (25 μM) was titrated into PP1 (3–4 μM) using an Affinity SV ITC micro-calorimeter at 25 °C (TA Instrument). Data were analysed using NITPIC, SEDPHAT and GUSSI.

**Bio-layer interferometry assay**. Bio-layer interferometry (BLI) is an optical analytical technique for measuring kinetics of interactions in real-time. The biosensor tip surface immobilized with a ligand is incubated with an analyte in solution, resulting in an increase in optical thickness at the biosensor tip and a wavelength shift, which is a direct measure of the change in thickness. Biolayer interferometry analysis of ASPP bindind to PP1 were studied using Octet Red 96 (ForteBio). In total, 50 μg/ml HIS tagged PPα, PP1β or PP1γ1 were immobilized on Nickel coated biosensor (Ni-NTA, ForteBio) and the typical immobilization levels were above 3 nm. Ligands-loaded Ni-NTA biosensors were then incubated with different concentrations of ASPP in the kinetics buffer. Global fitting of the binding curves generated a best fit with the 1:1 model and the kinetic association and dissociation constants were calculated. All binding experiments were performed in solid-black 96-well plates containing 200 μL of solution in each well at 25 °C with an agitation speed of 1000 rpm. Curve fitting, steady state analysis, and calculation of kinetic parameters ($K_{on}$, $K_{off}$ and $K_D$) were done using Octet software version 7.0 (ForteBio).

**Expression and purification of PP1α[301-330] for NMR**. Human PP1α[301-330], was cloned into the pET-M30-MBP plasmid that encodes an N-terminal His₆-tag followed by maltose binding protein (MBP) and a TEV (tobacco etch virus) protease cleavage site and expressed in *E. coli* BL21 (DE3) CodonPlus-RIL cells (Agilent Technologies). Cells were grown in Lysogeny Broth in the presence of selective antibiotics at 37 °C up to OD₆₀₀ of 0.6–0.8, and expression was induced by the addition of 1 mM isopropyl-β-D-1-thiogalactopyranoside (IPTG). Proteins were expressed for ~14 h at 18 °C prior to harvesting by centrifugation at 6,000 × g. Cell pellets were stored at −80 °C.

Cell pellets were suspended in lysis buffer (20 mM Tris-HCl pH 8.0, 500 mM NaCl, 5 mM Imidazole, 0.1% Triton X-100; EDTA-free protease inhibitor, [Roche]), and lysed using high-pressure homogenization (Avestin C3 EmulsiFlex). The cell lysate was centrifuged at 45,500 × g and the soluble fraction was loaded onto 10 ml of pre-equilibrated Ni-NTA resin (GE Healthcare). After 1 h at 4 °C, the resin was washed with five column volumes of wash buffer (20 mM Tris-HCl pH 8.0, 500 mM NaCl, 10 mM imidazole) and eluted with 20 ml of elution buffer (20 mM Tris-HCl pH 8.0, 500 mM NaCl, 500 mM imidazole). The purified protein was cleaved overnight at 4 °C with tobacco etch viral (TEV) protease and then subjected to a Ni-NTA purification step to remove the cleaved His₆-tagged MBP, any uncleaved protein and TEV protease. Purified PP1α[301-330] was pooled and heat purified (95 °C, 30 min; centrifuged at 12,000 × g) and the supernatant concentrated and immediately frozen and stored at −20 °C.

For NMR measurements, expression of uniformly ¹⁵N- and/or ¹³C-labelled PP1α[301-330] was facilitated by growing cells in M9 minimal media containing 1 g/l ¹⁵NH₄Cl and/or 4 g/l [¹³C]-*D*-glucose (CIL) as the sole nitrogen and carbon sources, respectively.

**NMR spectroscopy**. The sequence-specific backbone assignment of ¹⁵N,¹³C-labelled PP1α[301-330] was achieved by series of 3D NMR spectra including 3D HNCACB, 3D CBCA(CO)NH, 3D HNCA and 3D HN(CO)CA. All the spectra for PP1α[301-330] were obtained at 283 K using a Bruker Neo 600 MHz spectrometer equipped with a TCI HCN z-gradient cryoprobe. NMR spectra were processed with NMRPipe or Topspin 4.0.1 (Bruker) and analysed using either CARA (http://www.cara.nmr.ch) or Sparky (http://www.cgl.ucsf.edu/home/sparky).

¹H, ¹⁵N and ¹³C chemical shifts were indirectly referenced to 3-trimethyl-sylil-1-propanesulfonic acid, sodium salt (DSS). Chemical shift index (CSI) and secondary structure propensity (SSP) were calculated using the RefDB[76] database. CSI (deviations of chemical shifts from their expected random coil values, also known as secondary chemical shifts) was determined using the formula: CSI = $\Delta\delta C^\alpha - \Delta\delta C^\beta$, where $\Delta\delta C^\alpha = \delta_{observed}C^\alpha - \delta_{RefDB}C^\alpha$ and $\Delta\delta C^\beta = \delta_{observed}C^\beta - \delta_{RefDB}C^\beta$. SSP scores were calculated using ¹³Cᵅ and ¹³Cᵝ chemical shifts, five-residue weighted averaging and default parameters, using the RefDB programme. Cysteine and residues immediately preceding proline were ignored in secondary chemical shift calculations[77].

PP1α[301-330] and PP1α[301-330] saturated with ASPP2 (twofold excess) steady state heteronuclear ¹⁵N[¹H] NOE spectra (128 scans) were obtained using a 5 s recycle delay (with and without presaturation). Additionally, ¹⁵N transverse relaxation (CPMG) measurements (64 scans) were recorded using relaxation delays (16.96, 33.92, 67.84, 135.68, 203.52, 271.36, 339.2, 407.04 and 542.72 ms for PP1α[301-330]; 16.96, 33.92, 50.88, 67.88, 84.8, 101.76 and 118.72 ms for PP1α[301-330] saturated with ASPP2) using a 3 s recycle delay.

**Statistics**. Statistical analysis was performed using Graphpad Prism 7. For comparisons between two samples, two-tailed Students *t*-test were used. For multiple comparisons, one-way analysis of variance with Bonferroni correction was used. All graphs show the mean with the standard deviation of the mean and raw data points are indicated. Statistical significance is indicated by asterisks. See Supplementary table 3 for statistic analysis.

**Reporting summary**. Further information on experimental design is available in the Nature Research Reporting Summary linked to this article.

## Data availability
Data supporting the findings of this manuscript are available from the corresponding authors upon reasonable request. A reporting summary for this Article is available as a Supplementary Information file. The NMR chemical shifts have been deposited in the BioMagResBank, www.bmrb.wisc.edu (accession no. 27464). The atomic coordinates and structure factors have been deposited in the Protein Data Bank, www.pdb.org (PDB ID codes 6GHM). The mass spectrometry proteomics data have been deposited to the ProteomeXchange Consortium via the PRIDE partner repository with the dataset identifier PXD012378.

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

## Acknowledgements

We thank the Francis Crick Institute fermentation science technology platform for their help during protein expression, the Structural Biology science technology platform, the Crick Fly Facility, the Crick Advanced Light Microscopy facility for their technical support, A. Snijders and S. Howell from the Crick Proteomics science technology platform and Diamond Light Source for access to beamline I02 (mx9826-26) that contributed to the results presented here. We thank N. Selevsek from the Functional Genomic Center Zürich for mass spectrometer maintenance and M. Mehnert from the Gstaiger lab for molecular biology support. We are grateful to X. Lu, M. Glover and C. Holmes for reagents and sharing data prior to publication. We thank P. Ribeiro for comments on the manuscript. This work was supported by the Francis Crick Institute, which receives its core funding from Cancer Research UK (FC001175), the UK Medical Research Council (FC001175), and the Wellcome Trust (FC001175), as well as a Wellcome Trust Investigator award (107885/Z/15/Z). Work in the Peti and Page labs is supported by the NIH (R01-GM098482 to R.P. and R01-NS091336 to W.P.). Research in the Gstaiger lab is supported by the European Union 7th Framework Programme SYBILLA (Systems Biology of T-Cell Activation) and the Innovative Medicines Initiative project ULTRA- DD (grant agreement n° 115766).

## Author contributions

S.M. and R.L. solved the crystal structure and measured ITC affinities, S.M. performed the BLI measurements. Y.Z., M.T.B. and J.B. performed all cell culture and *Drosophila* experiments and performed molecular cloning for the structural and biophysical studies. N.O'R. synthesized peptides. R.B. performed the NMR studies; G.S.K. measured ITC affinities of full-length PP1. F.U., A.vD., S.H. and M.G. provided Mass Spectrometry data. N.T., S.M., M.G., R.P. and W.P. supervised the study. All authors contributed to manuscript writing.

## Additional information

**Competing interests:** The authors declare no competing interests.

