## [Peer Review File · Nature Communications]

Reviewers' Comments:

Reviewer #1:

Remarks to the Author:

The manuscript by Zhou et al describes novel structural, biophysical and biochemical studies that decipher the association of ASPP proteins, specifically ASPP2, with the protein phosphatase-1 (PP1) catalytic subunit with the key focus being on the preferential recognition of the PP1 α isoform by ASPP2 and other ASPP proteins. The work has been well carried out and the data in large part support the conclusions reached. While the analyses of the ASPP2/PP1 complex is quite thorough, the functional impact is largely assessed using genetic deletion or mutation of individual components and provide little or no insights into the function (specifically putative phosphoprotein substrates such as p53) for the ASPP2/PP1 α complex. Thus, having only examined two main endpoints, namely rough eye phenotype and bristle displacement, it might be a bit too presumptuous or premature to state (see abstract) "that PP1 association is necessary for all of *Drosophila* ASPP's in vivo functions". In any case, the manuscript is quite lengthy and in places, as shown above, somewhat speculative. Reducing some text and eliminating speculations for which the authors have no direct data or relate to issues somewhat peripheral to the work described (see specific comments), would provide a much tighter focus and potentially, enhance the impact of this work.

Specific comments :

1. The opening paragraph in the introduction makes the comparison of the two protein modifications, namely phosphorylation and ubiquitylation. This seems like an unnecessary distraction as protein ubiquitylation is not even considered or investigated in this work. Page 3 – "four PP1 catalytic subunits encoded by three loci exist" – If the authors mean "genetic loci" or simply "genes" then using the word "genes" may be clearer to the average reader. Page 5 – the transition between the paragraph ending with the statement that ASPP2 and *Drosophila* ASPP being localized "at tight junctions and adherens junctions" to that describing the function of ASPP proteins in cell cycle or gene expression seems somewhat sudden and difficult to relate and certainly gives no clues as to why PP1 isoform preference was the first order of priority for these studies.
2. Results – page 7 – While assessing several different components of the proposed ASPP complex, including PP1, RASSF8 and Ccdc85 was an interesting starting point, there are no data presented in support the presence of PP1 interacting SILK motif in Ccdc85. Given the relative shortness of this motif and even that of the more prevalent RVxF motif analyzed in ASPP, it would highly unreliable to speculate using bioinformatics alone that "*Drosophila* Ccdc85 possesses a SILK motif that is not conserved in mammalian CCDC85 proteins". This is after all the strength of structural studies for defining bona fide PP1-interacting SLiMs. Regardless, this statement does not belong in "Results".
3. The authors do not specifically comment on the distinct mobilities of *Drosophila* PP1 isoforms, 96A and 9C, that bind ASPP versus those that do not, namely 13C and 7B (Figures 1B, 4B and 4C). To what extent is this due to covalent modification of PP1, such as the PPII sequence that binds SH3 domain of ASPP or their differing C-terminal sequences?
4. Results – page 7 – "These experiments revealed that both RASSF8 and Ccdc85 interact with the N-terminal coiled-coil region of ASPP" may be an overstatement as IP studies, while useful in identifying potential components of multiprotein complexes, these assays may not by themselves define binding interactions. At best the authors can say that "the coiled-coil domain was required for the recruitment of both RASSF8 and Ccdc85 to the ASPP/PP1 complex in *Drosophila* S2 cells".
5. In the section describing the eye phenotypes of Ccdc85, RASSF8 and ASPP mutant flies showed some similarities and some differences but most importantly, these do not directly evoke a role for PP1 in these outcomes – as acknowledged by the authors (page 8, line 214). The fact that ASPPFA, a PP1 non-binding mutant, rescued albeit only partially compared to WT ASPP further emphasizes that there may be non-PP1-mediated impact of ASPP deletion in the phenotypes described for the mutant flies (see comment above regarding abstract). Here again, one would

advise some moderation of language in that "association of PP1 is required for (some or many) the in vivo functions of ASPP" (page 9 line 235).

6. Page 10, line 261 – "first example of a PIP that engages the PPII motif of the PP1 C-tail". As this involves the SH3 domain in ASPP, one wonders to what extent the engagement of PP1 C-tail micks or differs from other associations with the SH3 domain of ASPP or other proteins that have been analyzed by structural biology. Said differently, is there any evidence that the presence of other PP1-binding SLiMs allows a docking of PP1 C-tail (which appears remarkably low affinity i.e. 5-20 micromolar – page 10, line 298) in a manner different from previously analyzed target interactions with SH3 domains?

7. Page 11 paragraph starting line 303 – This paragraph is a little difficult to follow. As the core PPII sequence shared among all PP1 isoforms shows a "much lower affinity of 3200 nM", which is increased when the sequence is extended by seven residues to 270 nM, what does that reveal about isoform specificity. While the discussion focuses on "a critical role of K327, K329 and K330 in PP1 in providing additional affinity and specificity towards ASPP2 SH3 domain" the authors do not comment further on the specific differences between PP1alpha and other isoforms in this region and how that might impact isoform selectivity. On Page 16, line 479, the authors state that there is "The variability observed in this region for the other PP1 isoforms results in reduced affinity" but it is unclear from Figure 6 that which specific residues are accredited with this variability and whether this was enough to account for all the selectivity seen with different PP1 isoforms.

8. Page 13, paragraph starting line 370 is also confusing – specifically "though human iASPP binds the PP1alpha C-tail with relatively high affinity (430 nM), the binding is barely affected by the PP1alpha 3KA mutation (690 nM), showing that the human iASPP specificity pocket...does not interact with the PP1alpha C-terminal lysines". Is this not inferred rather than proven? Is it not also likely that the "more stable complex" of iASPP/PP1 due to other interactions is much less reliant on the association through three lysines. This paragraph as a whole could use some modification/moderation of text.

9. Page 15, some more confusing statements – on line 434, the authors state that the mutation of the SH3 specificity pocket had not discernable impact on bristle misplacements especially as it reduced binding to PP1. Yet in the second paragraph where the authors state "since *Drosophila* Ccdc85 has a PP1-binding SILK motif" (is this information derived from other work as not analyzed here?)...mutation of ASPP and ccdc85 (this looks like a gene deletion rather than a point mutation) together dramatically increased IOCs and the bristle misplacement phenotypes". These defects could be rescued back to ccdc85 single (allele) mutant levels by expression of WT ASPP. In contrast, all the ASPP mutant forms failed... These results show that both the RVxF and the SH3 domain are important for ASPP function in vivo". I can't for the life of me see the link between the putative SILK motif in Ccdc85 and the final statement regarding RVxF and SH3 regions in ASPP.

10. Page 16 lines 464 and 468..."Ccdc85/PP1 interaction is likely to be direct since fly Ccdc85 has a SILK SLiM"....but in mammalian cells where Ccdc85 also associates with PP1 (but does have the putative SILK motif), whether this interaction is direct is not clear". This either says that SILK is not required or that IPs do not provide direct evidence of binding sites.

11. Page 17, paragraph starting line 492 – seems highly speculative and something of a tangent – shorten or delete.

12. Page 18, lines 525 onwards...is the speculation about p53 binding to iASPP warranted? No analysis of p53 phosphorylation or dephosphorylation is presented in this manuscript.

13. Page 19 - Given that PPII domain in PP1 is known to be phosphorylated, it is surprising that authors did not evaluate the impact of eliminating this modification on ASPP2 association and/or function.

14. The final sentence regarding GADD34 is misleading as work by one of the authors has shown very clearly that GADD34/PP1 complex does not bind at least one of the drugs, Guanabenz, that has been postulated to inhibit or disrupt this phosphatase complex and at least two very detailed biochemical studies from David Ron's lab show that this drug does not influence the assembly or activity of the GADD34-containing eIF2alpha phosphatase. Thus this sentence should be deleted.

Reviewer #2:

Remarks to the Author:

In the present manuscript Zhou Yanxiang et al. describe how ASPP proteins interact with the protein phosphatase 1 (PP1) catalytic subunit through different sequence motifs. They provide a very comprehensive and data rich study combining in vitro and in vivo approaches. The authors suggest an important role for the C-terminal tail of PP1 in binding to the SH3 domain of ASPP proteins. While most of the experimental data are of good quality it feels that the cell culture data are not well integrated with the in vitro data and vice versa. Initially the authors verify the interactions between PP1 and ASPP by pull-downs, genetics, rescue assays and fluorescence microscopy. Furthermore they can show that the PP1/ASPP complex interacts with RASSF8 and Ccdc85. This section is followed by an in depth biophysical and structural characterization of the PP1 and ASPP interaction, but it is not extended to characterize the interaction to the other proteins. A number of PP1-PIP complex structures are already known and crucial sequence motifs have been identified (incl. the C-terminal tail of ASPP proteins; see Skene-Arnold TD, 2013). The present study reports about the important contribution of the C-terminal tail of PP1 to the interaction with ASPP proteins.

There are a number of comments and questions which need further clarification:

1.) The important C-terminal residues are missing in the electron density of the complex structure. Do the analyzed crystals contain full length PP1 (7-330) or can degradation be a problem? Did the authors try to crystallize ASPP2 with the C-terminal peptide of PP1?

2.) Although the affinity of the C-terminus of PP1 to ASPP2 is quite high and this is one of the major findings in the manuscript, the quality of the X-ray data is in this area somewhat limited as evident from the validation report (residues not modeled, poor fit to the electron density).

3.) The authors mainly used ITC and BLI to characterize the interaction between PP1 and ASPP members.

- Is there a specific reason why ITC was used only for probing the interaction of ASPP proteins with the C-terminal peptide of PP1 and not in the full length context as done in previous publications? Although I'm aware that this requires more material, it would allow to compare resulting affinities between the different techniques more efficiently (I would recommend to provide ITC data for ASPP2 interaction with PP1 (7-330) and PP1 (7-300) and placing the resulting data in the context of the BLI results.

- Please provide fits to the BLI data.

- Please provide information on how the concentrations of the C-terminal peptides of PP1 were determined since they do not contain any aromatic residues for absorption determination. This could be one reason why the stoichiometry for a number of ITC measurements is quite off (e.g. see S5D,F, K, M) which can have an effect on the resulting affinity. ITC data do not only provide information on the affinity but also allow to dissect the enthalpy and entropic contribution of the binding event. I believe, analyzing the ITC data in more detail would give more insights into the binding mode also regarding the proposed "fuzzy" charge-charge interaction. (Alternatively an interaction study at different pHs values could give valuable insights)

4.) NMR spectroscopy is a powerful tool to monitor dynamics and map interaction surfaces. 97% of the backbone resonances of the C-terminal peptide were assigned.

- why are hNOE and T2 values for a number of residues (in the apo form) missing (except for Pro)?

- the authors state on page 14 l.409 that residues 310-325 disappear upon addition of ASPP2: How is it possible to determine hNOE and T2 values for some of the residues shown in Figure 7D,E?

- To visualize the changes in the NMR spectra, I would recommend to provide "zoom ins" for Fig. 7 and S8D

5.) Title of the manuscript: I would recommend not to use the word "combinatorial"

Zhou Yanxiang et al. present a very data rich study on the interaction of PP1 to ASPP proteins. At the current stage it feels little effort has been made to integrate the cell biology data with the biophysical data and for the latter a more detailed analysis of the binding data is required.

Reviewer #3:

Remarks to the Author:

The authors investigate the interaction of PP1 isoforms with ASPP proteins. Starting in drosophila, they noted a dependence of binding to PP1 isoforms on their C-terminal tail, which in drosophila is truncated in two of the four isoforms. In human, the C-terminal tail is different in the PP1 isoforms and not truncated. Previously, it was reported that the different C-tails are important for the isoform-specificity of binding of PP1gamma2 against endophilin-B1t (ref 60). Also the interaction of human PP1 isoforms with different ASPP proteins was studied in vitro before, where the SH3 domain of ASPP proteins were found to bind to the C-tail type-2 SH3 (Src homology 3) poly-proline motif (PxxPxR) in PP1 isoforms (Ref 29). This paper did not report on isoform specificity. This is the major novelty of the current manuscript. Therefore, it should be solidly presented. The authors conduct thorough structural studies on PP1alpha, and present the first structure on the C-tail interacting with a PP1-interacting protein. They carry out ITC studies and NMR studies to investigate the binding in more detail, and conclude that the differences C-terminal of the poly-proline motif provide isoform specificity (hydrophobic/charge interactions). Then, they go back and investigate the interaction in vivo, in drosophila, where the relevance of this C-terminal stretch-induced specificity is not given because only two isoform have it and these bind ASPP equally well according to the IP experiments in Fig. 4. Also, if I understand correctly there is only one ASPP protein in drosophila, therefore the differences between the ASPP proteins in binding to isoforms of PP1 in human can also not be matched in drosophila. This brings me to my major concern:

The relevance of the selectivity measured between human PP1 isoform-tails and ASPP proteins is unclear. The selectivity is rather weak (6 fold, 8 fold). Does this play a role in vivo? Especially, since the authors show that a protein (CCDC85) can help rescue mutations of the PP1 C-tail in drosophila, could this be the case in human as well (maybe not via CCDC85, as the human proteins do not have the SILK motif)? If so, would this not mask the selectivity? I think that the relevance has to be addressed experimentally, for example using co-IPs or pull-downs of PP1 isoforms in HEK cells and identifying bound proteins and investigating their SH3 domains and adjacent acidic pockets computationally, or structural investigations with PP1beta or gamma1. Ideally, functional consequences of this specificity would be shown in human cells, not only effects on binding. To me this is crucial for the publication to be of broad relevance, as many of the findings were already presented before. If the selectivity cannot be proven relevant in human cells then the only novelty is the above-mentioned structure, which is on its own not of broad interest.

Minor concerns:

- 1) p.9 lines 235-236: "association of PP1 is required for the in vivo functions of ASPP." This is a strong statement. I am aware that the data is often interpreted that way, but if other proteins could bind to this motif in ASPP, which would be disrupted by the mutation, then this is no proof that the interaction with PP1 is the relevant one for the phenotypes. Is there for example MS-data that shows that no other protein binding to ASPP is affected by this mutation?
- 2) The authors sometimes confuse the reader with the order of the figures (for example, in the text fig 5C is mentioned before 5B etc).
- 3) P. 19 line 565: It is surprising that the authors use this example (GADD34) for a successful PP1 regulatory subunit targeting, when one of the authors has just shown that the holoenzyme is not effected by the drug (JBC 2018), and the subject is still under strong debate.

In conclusion, I think that the work has the potential for strong impact, if the biological relevance of the selectivity in human cells can be shown thoroughly. The manuscript is well written and carefully prepared.

Reviewer #4:

Remarks to the Author:

The manuscript by Zhou and colleagues provides important structural insight to understand how specificity in PP1-PIPs association is achieved. Resorting to biochemical and structural analysis the authors propose that ASPP proteins (regulatory/adaptor subunits of PP1) discriminate between the different PP1 isoforms through an acidic specificity pocket in the SH3 domain that displays differential affinity towards PP1 C-terminal tails. The reported findings are globally solid and contribute with important knowledge to better comprehend the still elusive PP1 regulatory code. Thus, the work is expected to be of broad interest to those in many different fields of biology, making it particularly appropriate for publication in Nature Communications.

However, a few questions/points should be addressed:

Major points:

1) Mutations of the ASPP RVxF (FA) or of the hydrophobic patch in the SH3 domain (WK) alone reduce binding to PP196 and PP19C in co-IP experiments. Does a double mutation (FA+WK) compromise further the binding to PP1, thus revealing a synergistic or additive contribution of these regions for the interaction with PP1? This would correlate nicely with the *in vivo* data presented in Fig S9 and argue in favour of multiple motifs increasing the binding affinity for PP1.

2) Can the authors demonstrate *in vivo* to which extent is the interaction of PP1 with the ASPP-FA, WK and KVK mutants (alone and in combination with *ccdc85*^{-/-}) compromised? ASPP localizes at adherens junctions. Do these mutations preclude PP196A and/or PP19C localization at these structures? Is the dephosphorylation of ASPP-PP1 substrates affected in these mutants?

Minor points:

1) The authors claim in the abstract that PP1 association with *Drosophila* ASPP is required for ALL *in vivo* functions (line 39) of the latter. This feels like an overstatement. It should be stated that PP1 association is required for *Drosophila* ASPP functions under study.

2) The authors demonstrate in the present manuscript that *ccdc85* mutants phenocopy the ASPP mutant eye phenotype. This resembles the phenotype of RASSF8 mutants that had been previously described by the same lab, as referenced in the text. Although I agree with the conclusion, no RASSF8 mutants were analysed in the present study. It seems more accurate to report in the title of the RESULTS section "*ccd85* mutants phenocopy the ASPP mutant eye phenotype" (line 187).

3) In the graphs of Fig 2F and 2G, *GMR>cc85* should be replaced by *GMR>ccdc85*.

4) In the graphs of Fig 2L and 2M it should be explicit that the transgenes ASPP-WT and ASPP-FA are expressed in a ASPP-null background.

5) The overlay in Figure 3L only displays the *MS1096>ASPP-WT* genotype. The *MS1096>ASPP-FA* seems to be missing.

Summary of new results:

1. We performed new ITC experiments showing that the PP1 α C-tail is required for interaction with ASPP2 in the context of full-length PP1 (Figure 6A), backing up our previous data using PP1 C-tail peptides.
2. By carrying out more ITC experiments with mutated PP1 α C-tail peptides, we directly test the contribution of individual lysines of the C-tail for the ASPP2/PP1 α interaction (Figure 6C).
3. We use an unbiased quantitative pulldown/Mass Spectrometry approach from HEK293 cell lysates to test our *in vitro* affinity data in a cellular context:
 - We show that ASPP2 is strongly associated with PP1 α and the ASPP2^{KVK} mutation, which disrupts the interaction between the ASPP2 SH3 domain and the PP1 C-tail *in vitro*, disrupts the association with PP1 α without affecting other ASPP2 partners (Figure 8A, S9). In contrast, the weaker association of ASPP2 with PP1 β and PP1 γ 1 is not affected by the ASPP2^{KVK} mutation. This suggests that the ASPP2 SH3 domain specificity pocket is indeed required for PP1 α recruitment in a cellular context.
 - We find that truncation of the PP1 α C-tail results in loss of binding to all ASPP/PP1 complex members (RASSF7/8, CCDC85 and ASPP2) to PP1 α (Figure 8C). This clearly implicates the PP1 C-tail in assembly of the ASPP/PP1 tetrameric complex.
 - We compare the association of ASPP1/2 with different PP1 subunit and find that, unlike ASPP1, ASPP2 shows marked selectivity towards with PP1 α versus PP1 β and PP1 γ 1 (Figure 8B).
4. Using the known ASPP/PP1 substrate TAZ (a transcriptional co-activator target of the Hippo tumour-suppressor pathway), we show that, similar to mutations of the RVxF motif, the ASPP2^{KVK} mutation in the SH3 domain specificity pocket prevents ASPP2 from promoting TAZ dephosphorylation on Serine 89 in cell culture (Figure 8D, E).
5. We have analyzed the effect of a second mutation affecting the *Drosophila* ASPP SH3 domain (ASPP^{WK}) on eye development and find it also leads to disrupted retinal morphogenesis (Figure 9G, I, J). Thus, two independent mutations in the ASPP SH3 domain (ASPP^{WK} and ASPP^{KVK}) that disrupt PP1 binding *in vitro* and in co-IPs reduce ASPP function *in vivo*, further supporting our model that PP1 association via the SH3 domain is required for ASPP function.
6. Our previous biochemical data had shown that *Drosophila* ASPP interacts robustly with PP1 α 96A and PP1 β 9C, which have a C-tail, and only weakly with PP1 α 87C and PP1 α 13C, which lack a C-tail (Figure 1B). We examine the localization of PP1 isoforms *in vivo* in the *Drosophila* retina and identified a pool of PP1 α 96A and PP1 β 9C (also known as Flap wing – Flw) localized at the adherens junctions (Figure S10J-K'), where ASPP and RASSF8 are localized (Langton et al Curr Biol 2009), whereas PP1 α 87C shows no junctional enrichment (Figure S10I, I'). Interestingly, disrupting the ASPP/PP1 complex by loss of ASPP and Ccdc85 causes the loss of the junctional pools of PP1 α 96A and PP1 β 9C (Figures 9K-N and S10L-N). This shows that the ASPP complex is required to direct the subcellular localization of PP1 isoforms *in vivo*.

Thus, our new data strengthen the key points of our manuscript:

- ASPP2/*Drosophila* ASPP nucleate PP1 complexes and the association with PP1 is required for normal ASPP function *in vivo*.

- ASPP proteins recruit PP1 using a canonical RVxF motif and their SH3 domain, which binds the divergent PP1 C-tail.
- ASPP2 and *Drosophila* ASPP can discriminate between PP1 isoforms based on their C-tails. This recognition involves both an SH3/polyproline motif interaction and the SH3 domain specificity pocket, which interacts with basic residues in the PP1 α C-tail.

Detailed response to the reviewers' comments

Reviewer #1 (Remarks to the Author):

The manuscript by Zhou et al describes novel structural, biophysical and biochemical studies that decipher the association of ASPP proteins, specifically ASPP2, with the protein phosphatase-1 (PP1) catalytic subunit with the key focus being on the preferential recognition of the PP1 α isoform by ASPP2 and other ASPP proteins. The work has been well carried out and the data in large part support the conclusions reached. While the analyses of the ASPP2/PP1 complex is quite thorough, the functional impact is largely assessed using genetic deletion or mutation of individual components and provide little or no insights into the function (specifically putative phosphoprotein substrates such as p53) for the ASPP2/PP1 α complex. Thus, having only examined two main endpoints, namely rough eye phenotype and bristle displacement, it might be a bit to presumptuous or premature to state (see abstract) “that PP1 association is necessary for all of *Drosophila* ASPP’s *in vivo* functions”. In any case, the manuscript is quite lengthy and in places, as shown above, somewhat speculative. Reducing some text and eliminating speculations for which the authors have no direct data or relate to issues somewhat peripheral to the work described (see specific comments), would provide a much tighter focus and potentially, enhance the impact of this work.

We thank the reviewer for his/her useful comments. As detailed above, we have added several new pieces of functional data linking our structural insights with function. In particular, we show that the ASPP2/PP1 interaction is necessary for dephosphorylation of a known substrate, the Hippo pathway transcriptional co-activator Taz (Figure 8D, E). Furthermore, we show that the ASPP/PP1 complex components ASPP and Ccdc85 are required for junctional localization of two PP1 isoforms in the *Drosophila* retina (Figures 9K-N and S10L-N). We have nevertheless toned down the statement about PP1 interaction in the abstract (“We show that *Drosophila* ASPP is part of a multiprotein PP1 complex and that PP1 association is necessary for several *in vivo* functions of *Drosophila* ASPP.”). As suggested by the reviewer, we have also trimmed the manuscript and removed speculative statements.

Specific comments :

1. The opening paragraph in the introduction makes the comparison of the two protein modifications, namely phosphorylation and ubiquitylation. This seems like an unnecessary distraction as protein ubiquitylation is not even considered or investigated in this work. Page 3 – “four PP1 catalytic subunits encoded by three loci exist” – If the authors mean “genetic loci” or simply “genes” then using the word “genes” may be clearer to the average reader. Page 5 – the transition between the paragraph ending with the statement that ASPP2 and *Drosophila* ASPP being localized “at tight junctions and adherens junctions” to that describing the function of ASPP proteins in cell cycle or gene expression seems somewhat sudden and difficult to relate and certainly gives no clues as to why PP1 isoform preference was the first order of priority for these studies.

- As requested, we trimmed the first paragraph by removing the ubiquitin part.
- p3 We have changed genetic loci for genes in the text as requested.
- p5: we have added text linking both paragraphs to help the reader.

“Indeed, mammalian ASPP2 and *Drosophila* ASPP localize at tight junctions and adherens junctions (AJs) respectively³³⁻³⁵ and are required for junctional stability, at least in part by recruiting the polarity protein Par-3 (Bazooka in flies)^{33,34,36}, although the role of the ASPP/PP1 association has not been examined in this context.”

2. Results – page 7 – While assessing several different components of the proposed ASPP complex, including PP1, RASSF8 and Ccdc85 was an interesting starting point, there are no data presented in support the presence of PP1 interacting SILK motif in Ccdc85. Given the relative shortness of this motif and even that of the more prevalent RVxF motif analyzed in ASPP, it would highly unreliable to speculate using bioinformatics alone that “*Drosophila* Ccdc85 possesses a SILK motif that is not conserved in mammalian CCDC85 proteins”. This is after all the strength of structural studies for defining bona fide PP1-interacting SLiMs. Regardless, this statement does not belong in “Results”.

As suggested, we have removed this sentence.

3. The authors do not specifically comment on the distinct mobilities of *Drosophila* PP1 isoforms, 96A and 9C, that bind ASPP versus those that do not, namely 13C and 7B (Figures 1B, 4B and 4C). To what extent is this due to covalent modification of PP1, such as the PPII sequence that binds SH3 domain of ASPP or their differing C-terminal sequences?

The difference in mobility between *Drosophila* PP1 isoforms 96A and 9C vs 13C and 87B is due to the presence (96A and 9C) or absence (13C and 87B) of a C-tail (about 30 residues), see alignment in Figure 4A. In fact, as can be seen in Figure 4B, deletion of the C-terminal tail (Δ C) in both isoforms increases the mobility of these proteins to be comparable with 13C and 87B.

4. Results – page 7 – “These experiments revealed that both RASSF8 and Ccdc85 interact with the N-terminal coiled-coil region of ASPP” may be an overstatement as IP studies, while useful in identifying potential components of multiprotein complexes, these assays may not by themselves define binding interactions. At best the authors can say that “the coiled-coil domain was required for the recruitment of both RASSF8 and Ccdc85 to the ASPP/PP1 complex in *Drosophila* S2 cells”.

We have changed the text as requested.

5. In the section describing the eye phenotypes of Ccdc85, RASSF8 and ASPP mutant flies showed some similarities and some differences but most importantly, these do not directly evoke a role for PP1 in these outcomes – as acknowledged by the authors (page 8, line 214). The fact that ASPPFA, a PP1 non-binding mutant, rescued albeit only partially compared to WT ASPP further emphasizes that there may be non-PP1-mediated impact of ASPP deletion in the phenotypes described for the mutant flies (see comment above regarding abstract). Here again, one would advise some moderation of language in that “association of PP1 is required for (some or many) the in vivo functions of ASPP” (page 9 line 235).

We have changed the text as requested.

6. Page 10, line 261 – “first example of a PIP that engages the PPII motif of the PP1 C-tail”. As this involves the SH3 domain in ASPP, one wonders to what extent the engagement of PP1 C-tail micks or differs from other associations with the SH3 domain of ASPP or other proteins that have been analyzed by structural biology. Said differently, is there any evidence that the presence of other PP1-binding

SLiMs allows a docking of PP1 C-tail (which appears remarkably low affinity i.e. 5-20 micromolar – page 10, line 298) in a manner different from previously analyzed target interactions with SH3 domains?

The affinity we report for the ASPP2 interaction with the PP1 polyproline motif is typical for SH3 domain/polyproline motif interactions (in the 5-20uM range – Figure 6C and Mayer, BJ J Cell Sci 2001). Indeed, the crystal structure reveals that the PP1 polyproline binds to the ASPP2 SH3 domain in a classical fashion (Figure 5), comparable with published SH3 domain/ligand structures (Mayer, BJ J Cell Sci 2001). As the reviewer suggests, it is the combination of motifs (RVxF+SH3+acidic specificity pocket) that creates a high affinity ASPP/PP1 complex.

7. Page 11 paragraph starting line 303 – This paragraph is a little difficult to follow. As the core PPII sequence shared among all PP1 isoforms shows a “much lower affinity of 3200 nM”, which is increased when the sequence is extended by seven residues to 270 nM, what does that reveal about isoform specificity. While the discussion focuses on “a critical role of K327, K329 and K330 in PP1 in providing additional affinity and specificity towards ASPP2 SH3 domain” the authors do not comment further on the specific differences between PP1 α and other isoforms in this region and how that might impact isoform selectivity. On Page 16, line 479, the authors state that there is “The variability observed in this region for the other PP1 isoforms results in reduced affinity” but it is unclear from Figure 6 that which specific residues are accredited with this variability and whether this was enough to account for all the selectivity seen with different PP1 isoforms.

We thank the reviewer for raising this important point. Indeed, apart from PP1 γ 2, which only has one C-tail lysine, all PP1 isoforms have three lysines/arginines, therefore, the number of positively charged residues in the C-tail alone cannot explain the affinity difference (see alignment Figure 4A). We have performed further ITC experiments mutating the PP1 α C-tail lysines individually and in combination (Figure 6C). This reveals that the first lysine (PP1 α ^{K327}) has the most important in promoting a high affinity interaction (4x drop in affinity for this mutant compared to 2x for the other two lysines). Interestingly, PP1 α is the only isoform with a lysine at this position, the other isoforms have a longer spacer region between the polyproline and the lysines (Figure 4A). In addition, it is likely that the presence of a glycine or proline, which would affect C-tail geometry, has a negative effect on affinity. The PP1 β , PP1 γ 1 and PP1 γ 2 all have at least one glycine or proline, while PP1 α does not (Figure 4A). This view is supported by our chimera experiments where we combine the polyproline and C-tail extensions of the different isoforms (Figure 6C). This suggests that it is both the position and conformational context of the C-tail lysines that determines isoform selectivity by the ASPP2 SH3 domain. To make this point come across more clearly, we have added some text on p11-12 as suggested by the reviewer, and in the discussion (p18-19).

8. Page 13, paragraph starting line 370 is also confusing – specifically “though human iASPP binds the PP1 α C-tail with relatively high affinity (430 nM), the binding is barely affected by the PP1 α 3KA mutation (690 nM), showing that the human iASPP specificity pocket....does not interact with the PP1 α C-terminal lysines”. Is this not inferred rather than proven? Is it not also likely that the “more stable complex” of iASPP/PP1 due to other interactions is much less reliant on the association through three lysines. This paragraph as a whole could use some modification/moderation of text.

We agree with the reviewer and have toned down this paragraph. In particular, the last sentence reads as follows:

“In order to understand how iASPP interacts with PP1 α C-tails, we performed ITC measurement, showing that, though human iASPP binds the PP1 α C-tail with relatively high affinity (395 nM), the

binding is barely affected by the PP1 α 3KA mutation (690 nM), showing that the iASPP/PP1 interaction is much less reliant on the PP1 α C-terminal lysines and therefore likely involves other interactions. In support of this view, the iASPP SH3 domain specificity pocket is less acidic than that of ASPP2 (Figure S7A)."

9. Page 15, some more confusing statements – on line 434, the authors state that the mutation of the SH3 specificity pocket had not discernable impact on bristle misplacements especially as it reduced binding to PP1. Yet in the second paragraph where the authors state "since *Drosophila* Ccdc85 has a PP1-binding SILK motif" (is this information derived from other work as not analyzed here?)...mutation of ASPP and ccdc85 (this looks like a gene deletion rather than a point mutation) together dramatically increased IOCs and the bristle misplacement phenotypes". These defects could be rescued back to ccdc85 single (allele) mutant levels by expression of WT ASPP. In contrast, all the ASPP mutant forms failed... These results show that both the RVxF and the SH3 domain are important for ASPP function *in vivo*". I can't for the life of me see the link between the putative SILK motif in Ccdc85 and the final statement regarding RVxF and SH3 regions in ASPP.

We agree with the reviewer that the reference to the SILK motif, the *in vivo* importance of which we have no evidence for, is not really necessary and is confusing. As for point 2 above, we have removed this reference.

10. Page 16 lines 464 and 468..."Ccdc85/PP1 interaction is likely to be direct since fly Ccdc85 has a SILK SLiM"....but in mammalian cells where Ccdc85 also associates with PP1 (but does have the putative SILK motif), whether this interaction is direct is not clear". This either says that SILK is not required or that IPs do not provide direct evidence of binding sites.

As discussed above, we have removed speculations concerning the SILK motif.

11. Page 17, paragraph starting line 492 – seems highly speculative and something of a tangent – shorten or delete.

As suggested by the reviewer, we have cut this paragraph and just leave one sentence to suggest other complexes between PP1 and SH3-domain containing proteins may exist.

12. Page 18, lines 525 onwards...is the speculation about p53 binding to iASPP warranted? No analysis of p53 phosphorylation or dephosphorylation is presented in this manuscript.

As suggested by the reviewer, we have removed the speculation about iASPP/p53 binding and phosphorylation.

13. Page 19 - Given that PPII domain in PP1 is known to be phosphorylated, it is surprising that authors did not evaluate the impact of eliminating this modification on ASPP2 association and/or function.

As there is little information on the functional significance or physiological relevance of PP1 C-tail phosphorylation, we have removed this speculative paragraph.

14. The final sentence regarding GADD34 is misleading as work by one of the authors has shown very clearly that GADD34/PP1 complex does not bind at least one of the drugs, Guanabenz, that has been postulated to inhibit or disrupt this phosphatase complex and at least two very detailed biochemical

studies from David Ron's lab show that this drug does not influence the assembly or activity of the GADD34-containing eIF2alpha phosphatase. Thus this sentence should be deleted.

The reviewer is correct. Guanabenz does not bind PP1 or GADD34; nor does it disrupt the GADD34-containing eIF2alpha phosphatase. Thus, we have deleted this sentence. However, there are data that shows that SLiM binding pockets are excellent targets for therapeutics. Namely, the immunosuppressant drugs FK506 and cyclosporin A bind the ser/thr phosphatase calcineurin at the LxVP binding pocket (the LxVP binding pocket is a CN-specific SLiM binding pocket). When they bind this pocket, they inhibit CN substrates, such as the NFATs, from binding CN, which in turn inhibits their dephosphorylation. We have added the following sentence to the manuscript:

"The ability to target specific SLiM interaction pockets in ser/thr phosphatases is supported by the discovery that the widely used immunosuppressants cyclosporin A and FK506 bind calcineurin in the LxVP SLiM binding pocket."

Reviewer #2 (Remarks to the Author):

In the present manuscript Zhou Yanxiang et al. describe how ASPP proteins interact with the protein phosphatase 1 (PP1) catalytic subunit through different sequence motifs. They provide a very comprehensive and data rich study combining in vitro and in vivo approaches. The authors suggest an important role for the C-terminal tail of PP1 in binding to the SH3 domain of ASPP proteins. While most of the experimental data are of good quality it feels that the cell culture data are not well integrated with the in vitro data and vice versa. Initially the authors verify the interactions between PP1 and ASPP by pull-downs, genetics, rescue assays and fluorescence microscopy. Furthermore they can show that the PP1/ASPP complex interacts with RASSF8 and Ccdc85. This section is followed by an in depth biophysical and structural characterization of the PP1 and ASPP interaction, but it is not extended to characterize the interaction to the other proteins. A number of PP1-PIP complex structures are already known and crucial sequence motifs have been identified (incl. the C-terminal tail of ASPP proteins; see Skene-Arnold TD, 2013). The present study reports about the important contribution of the C-terminal tail of PP1 to the interaction with ASPP proteins. There are a number of comments and questions which need further clarification:

1.) The important C-terminal residues are missing in the electron density of the complex structure. Do the analyzed crystals contain full length PP1 (7-330) or can degradation be a problem? Did the authors try to crystallize ASPP2 with the C-terminal peptide of PP1?

We used intact mass spectrometry in order to test whether any degradation might have occurred during the purification or crystallization process. We therefore purified the PP1a:ASPP2 complex and left it at 20°C for a few days to mimic the crystallization condition. After 7 days at 20°C, the mass spectrum indicates that both proteins forming the complex are intact (Figure S4C). This analysis clearly shows that PP1 is not degraded after 7 days at 20°C and therefore the missing electron density for PP1 C-tail is most likely not the result of a degradation problem.

We also tried to crystallize ASPP2⁹²⁰⁻¹¹²⁰ and ASPP2 SH3-domain with various peptide covering PP1α-Ctail but were unsuccessful.

However, our NMR (Figures 7 and S8), BLI and ITC (Figure 6) data clearly support the importance of the C-terminal residues in the ASPP2/PP1 α interaction.

2.) Although the affinity of the C-terminus of PP1 to ASPP2 is quite high and this is one of the major findings in the manuscript, the quality of the X-ray data is in this area somewhat limited as evident from the validation report (residues not modeled, poor fit to the electron density).

As discussed above, this is likely due to the flexibility of the ASPP2 N-Src loop and PP1 α C-tail, which interact together through “fuzzy” electrostatic interaction. This model is supported by our NMR (Figure 7), BLI and ITC data (Figure 6), as well as our *in vivo* analysis of *Drosophila* ASPP (Figure 9) and mammalian ASPP2 (Figure 8).

3.) The authors mainly used ITC and BLI to characterize the interaction between PP1 and ASPP members.

- Is there a specific reason why ITC was used only for probing the interaction of ASPP proteins with the C-terminal peptide of PP1 and not in the full length context as done in previous publications? Although I'm aware that this requires more material, it would allow to compare resulting affinities between the different techniques more efficiently (I would recommend to provide ITC data for ASPP2 interaction with PP1 (7-330) and PP1 (7-300) and placing the resulting data in the context of the BLI results.

As requested, we have performed ITC experiments using ASPP2 with both PP1 α (7-330) and PP1 α deleted of the C-terminus (7-300). The data show that the K_D of ASPP2 for PP1 α (7-330) is 6.1 ± 1.1 nM, while the K_D of ASPP2 for PP1 α (7-300) cannot be determined due to a lack of signal – confirming the BLI data. This data is now included in Figure 6A.

- Please provide fits to the BLI data.

As requested, these have been added to Figure 6B.

- Please provide information on how the concentrations of the C-terminal peptides of PP1 were determined since they do not contain any aromatic residues for absorption determination. This could be one reason why the stoichiometry for a number of ITC measurements is quite off (e.g. see S5D,F, K, M) which can have an effect on the resulting affinity. ITC data do not only provide information on the affinity but also allow to dissect the enthalpy and entropic contribution of the binding event. I believe, analyzing the ITC data in more detail would give more insights into the binding mode also regarding the proposed "fuzzy" charge-charge interaction. (Alternatively an interaction study at different pHs values could give valuable insights).

The peptide concentration was measured using peptide bond absorbance at 215nm on a Nanodrop Denovix spectrophotometer (peptide measurement method). As requested by the reviewer, we repeated those of the ITC measurement that showed a high molar ratio (Figures S5B, D, F, K and M in the previous submission). All the molar ratios are now within an acceptable range of 0.95 to 1.14 and the new data are consistent with our previous observations.

4.) NMR spectroscopy is a powerful tool to monitor dynamics and map interaction surfaces. 97% of the backbone resonances of the C-terminal peptide were assigned.

- why are hNOE and T2 values for a number of residues (in the apo form) missing (except for Pro)?

We thank the reviewer for her/his careful read and observation. The answer is overlap. Using 3D NMR data we can readily assign overlapped peaks, however we cannot distinguish peak intensities that are necessary for hNOE and T2 data. Overlapped peaks report on the combined intensities of the two overlapped peaks and thus cannot be used for these type of analysis. This leads to a lower number of data points – while obviously frustrating, there is no other way to do this accurately.

- the authors state on page 14 l.409 that residues 310-325 disappear upon addition of ASPP2: How is it possible to determine hNOE and T2 values for some of the residues shown in Figure 7D,E?

We thank the reviewer for highlighting this discrepancy. We used a much more concentrated sample and longer acquisition time to be able to see the residual intensities of the peaks that interact with ASPP2. Details are now provided in the methods section.

- To visualize the changes in the NMR spectra, I would recommend to provide “zoom ins” for Fig. 7 and S8D

We agree with the reviewer’s suggestion and have significantly updated both Fig. 7 and S8 to better display the NMR data; detailed versions of the spectra are now displayed in Figure 7.

5.) Title of the manuscript: I would recommend not to use the word “combinatorial”

We have changed the title to: ASPP proteins discriminate between PP1 catalytic subunits through their SH3 domain and the PP1 C-tail.

Zhou Yanxiang et al. present a very data rich study on the interaction of PP1 to ASPP proteins. At the current stage it feels little effort has been made to integrate the cell biology data with the biophysical data and for the latter a more detailed analysis of the binding data is required.

As discussed above, we have strengthened the binding data:

- ITC using full-length PP1 (Figure 6A).
- Improved peptide ITCs using better molar ratios (Figures 6C and S5).
- Clarification of the NMR data (see point 4 above).

In the revision, we have also performed a number of experiments to better integrate the biophysical data into a cellular context.

First, we have used unbiased Mass Spectrometry analysis to link the biophysical finding to the cellular context:

- We show that ASPP2 is strongly associated with PP1 α and the ASPP2^{KVK} mutation, which disrupts the interaction between the ASPP2 SH3 domain and the PP1 C-tail *in vitro*, disrupts the association with PP1 α without affecting other ASPP2 partners (Figure 8A, S9). In contrast, the weaker association of ASPP2 with PP1 β and PP1 γ 1 is not affected by the ASPP2^{KVK} mutation. This suggests that the ASPP2 SH3 domain specificity pocket is indeed required for PP1 α recruitment in a cellular context.
- We find that truncation of the PP1 α C-tail results in loss of binding to all ASPP/PP1 complex members (RASSF7/8, CCDC85 and ASPP2) to PP1 α (Figure 8C). This clearly implicates the PP1 C-tail in assembly of the ASPP/PP1 tetrameric complex.
- We compare the association of ASPP1/2 with different PP1 subunit and find that, unlike ASPP1, ASPP2 shows marked selectivity towards with PP1 α versus PP1 β and PP1 γ 1 (Figure 8B).

Secondly, we functionally link PP1 to ASPP function:

- Using the known ASPP/PP1 substrate TAZ (a transcriptional co-activator target of the Hippo tumour-suppressor pathway), we show that, similar to mutations of the RVxF motif, the ASPP2^{KVK} mutation in the SH3 domain specificity pocket prevents ASPP2 from promoting TAZ dephosphorylation on Serine 89 in cell culture (Figure 8D, E).
- We have analyzed the effect of a second mutation affecting the *Drosophila* ASPP SH3 domain (ASPP^{WK}) on eye development and find it also leads to disrupted retinal morphogenesis (Figure 9G, I, J). Thus, two independent mutations in the ASPP SH3 domain (ASPP^{WK} and ASPP^{KVK}) that disrupt PP1 binding *in vitro* and in co-IPs reduce ASPP function *in vivo*, further supporting our model that PP1 association via the SH3 domain is required for ASPP function.
- Our previous biochemical data had shown that *Drosophila* ASPP interacts robustly with PP1α96A and PP1β9C, which have a C-tail, and only weakly with PP1α87C and PP1α13C, which lack a C-tail (Figure 1B). We examine the localization of PP1 isoforms *in vivo* in the *Drosophila* retina and identified a pool of PP1α96A and PP1β9C (also known as Flap wing – Flw) localized at the adherens junctions (Figure S10J-K'), where ASPP and RASSF8 are localized (Langton et al Curr Biol 2009), whereas PP1α87C shows no junctional enrichment (Figure S10I, I'). Interestingly, disrupting the ASPP/PP1 complex by loss of ASPP and Ccdc85 causes the loss of the junctional pools of PP1α96A and PP1β9C (Figures 9K-N and S10L-N). This shows that the ASPP complex is required to direct the subcellular localization of PP1 isoforms *in vivo*.

Reviewer #3 (Remarks to the Author):

The authors investigate the interaction of PP1 isoforms with ASPP proteins. Starting in *Drosophila*, they noted a dependence of binding to PP1 isoforms on their C-terminal tail, which in *Drosophila* is truncated in two of the four isoforms. In human, the C-terminal tail is different in the PP1 isoforms and not truncated. Previously, it was reported that the different C-tails are important for the isoform-specificity of binding of PP1γ2 against endophilin-B1t (ref 60). Also the interaction of human PP1 isoforms with different ASPP proteins was studied *in vitro* before, where the SH3 domain of ASPP proteins were found to bind to the C-tail type-2 SH3 (Src homology 3) poly-proline motif (PxxPxR) in PP1 isoforms (Ref 29). This paper did not report on isoform specificity. This is the major novelty of the current manuscript. Therefore, it should be solidly presented. The authors conduct thorough structural studies on PP1α, and present the first structure on the C-tail interacting with a PP1-interacting protein. They carry out ITC studies and NMR studies to investigate the binding in more detail, and conclude that the differences C-terminal of the poly-proline motif provide isoform specificity (hydrophobic/charge interactions). Then, they go back and investigate the interaction *in vivo*, in *Drosophila*, where the relevance of this C-terminal stretch-induced specificity is not given because only two isoforms have it and these bind ASPP equally well according to the IP experiments in Fig. 4. Also, if I understand correctly there is only one ASPP protein in *Drosophila*, therefore the differences between the ASPP proteins in binding to isoforms of PP1 in human can also not be matched in *Drosophila*. This brings me to my major concern:

The relevance of the selectivity measured between human PP1 isoform-tails and ASPP proteins is unclear. The selectivity is rather weak (6 fold, 8 fold). Does this play a role *in vivo*?

Especially, since the authors show that a protein (CCDC85) can help rescue mutations of the PP1 C-tail in drosophila, could this be the case in human as well (maybe not via CCDC85, as the human proteins do not have the SILK motif)? If so, would this not mask the selectivity? I think that the relevance has to be addressed experimentally, for example using co-IPs or pull-downs of PP1 isoforms in HEK cells and identifying bound proteins and investigating their SH3 domains and adjacent acidic pockets computationally, or structural investigations with PP1beta or gamma1. Ideally, functional consequences of this specificity would be shown in human cells, not only effects on binding. To me this is crucial for the publication to be of broad relevance, as many of the findings were already presented before. If the selectivity cannot be proven relevant in human cells then the only novelty is the above-mentioned structure, which is on its own not of broad interest.

We have previously showed that a ~5-fold difference in K_D (*in vitro* ITC measurement) enables perfect isoform selectivity in cellular assays (fluorescent co-localization and targeting assays – Kumar, GS et al eLife 2016). We have now included a sentence which states this in the manuscript:

“Similar differences in affinity are sufficient to achieve PP1 isoform selectivity *in vivo*, as illustrated for RepoMan/Ki-67, which selectively bind to PP1 γ based on a single amino acid change in the PP1 catalytic domain¹⁵.”

In addition, we have performed several experiments to address the selectivity issue:

We use an unbiased quantitative pulldown/Mass Spectrometry approach from HEK293 cell lysates to test our *in vitro* affinity data in a cellular context:

- We show that ASPP2 is strongly associated with PP1 α and the ASPP2^{KVK} mutation, which disrupts the interaction between the ASPP2 SH3 domain and the PP1 C-tail *in vitro*, disrupts the association with PP1 α without affecting other ASPP2 partners (Figure 8A, S9). In contrast, the weaker association of ASPP2 with PP1 β and PP1 γ 1 is not affected by the ASPP2^{KVK} mutation. This suggests that the ASPP2 SH3 domain specificity pocket is indeed required for PP1 α recruitment in a cellular context.

- We compare the association of ASPP1/2 with different PP1 subunit and find that, unlike ASPP1, ASPP2 shows marked selectivity towards with PP1 α versus PP1 β and PP1 γ 1 (Figure 8B).

Finally, using the known ASPP/PP1 substrate TAZ (a transcriptional co-activator target of the Hippo tumour-suppressor pathway), we show that, similar to mutations of the RVxF motif, the ASPP2^{KVK} mutation in the SH3 domain specificity pocket prevents ASPP2 from promoting TAZ dephosphorylation on Serine 89 in cell culture (Figure 8xxx). Thus, the ASPP2 SH3 domain specificity pocket, which is required for selective binding to PP1 α is required for ASPP2/PP1 phosphatase function.

Minor concerns:

1) p.9 lines 235-236: “association of PP1 is required for the *in vivo* functions of ASPP.” This is a strong statement. I am aware that the data is often interpreted that way, but if other proteins could bind to this motif in ASPP, which would be disrupted by the mutation, then this is no proof that the interaction with PP1 is the relevant one for the phenotypes. Is there for example MS-data that shows that no other protein binding to ASPP is affected by this mutation?

As also requested by other reviewers, we have changed this sentence to “association with PP1 is required for several *in vivo* functions of ASPP” to denote the fact that PP1 might not be involved in all ASPP functions.

In addition, our new pulldown/Mass Spectrometry experiments indicate that the ASPP2^{KVK} mutation, which disrupts the ASPP2 SH3 domain specificity pocket, disrupts interaction with PP1 α , but not other ASPP2 interactors (Figure 8A, S9). Furthermore, we show *in vivo* that three independent mutations that target the ASPP/PP1 interaction (ASPP^{FA}, ASPP^{WK} and ASPP^{KVK}) all disrupt ASPP function *in vivo* (Figure 2K-M and 9A-J). In mammalian cells, we two independent mutations (ASPP2^{FA} and ASPP2^{KVK}) prevent dephosphorylation of TAZ (Figure 8D, E). Thus, it is highly unlikely that the phenotypes we observe are due to other ASPP/ASPP2 interactors.

We also showed that disruption of the ASPP complex *in vivo* leads to loss of junctional localization of two PP1 isoforms in the fly retina, further functionally linking ASPP with PP1 (Figures 9K-N and S10L-N).

2) The authors sometimes confuse the reader with the order of the figures (for example, in the text fig 5C is mentioned before 5B etc).

We have changed this as requested.

3) P. 19 line 565: It is surprising that the authors use this example (GADD34) for a successful PP1 regulatory subunit targeting, when one of the authors has just shown that the holoenzyme is not effected by the drug (JBC 2018), and the subject is still under strong debate.

The reviewer is correct. Guanabenz does not bind PP1 or GADD34; nor does it disrupt the GADD34-containing eIF2 α phosphatase. Thus, we have deleted this sentence. However, there are data that shows that SLiM binding pockets are excellent targets for therapeutics. Namely, the immunosuppressant drugs FK506 and cyclosporin A bind the ser/thr phosphatase calcineurin at the LxVP binding pocket (the LxVP binding pocket is a CN-specific SLiM binding pocket). When they bind this pocket, they inhibit CN substrates, such as the NFATs, from binding CN, which in turn inhibits there dephosphorylation. We have added the following sentence to the manuscript:

“The ability to target specific SLiM interaction pockets in ser/thr phosphatases is supported by the discovery that the widely used immunosuppressants cyclosporin A and FK506 bind calcineurin in the LxVP SLiM binding pocket.”

In conclusion, I think that the work has the potential for strong impact, if the biological relevance of the selectivity in human cells can be shown thoroughly. The manuscript is well written and carefully prepared.

Reviewer #4 (Remarks to the Author):

The manuscript by Zhou and colleagues provides important structural insight to understand how specificity in PP1-PIPs association is achieved. Resorting to biochemical and structural analysis the authors propose that ASPP proteins (regulatory/adaptor subunits of PP1) discriminate between the different PP1 isoforms through an acidic specificity pocket in the SH3 domain that displays differential affinity towards PP1 C-terminal tails.

The reported findings are globally solid and contribute with important knowledge to better comprehend the still elusive PP1 regulatory code. Thus, the work is expected to be of broad interest to those in many different fields of biology, making it particularly appropriate for publication in Nature Communications.

However, a few questions/points should be addressed:

Major points:

1) Mutations of the ASSP RVxF (FA) or of the hydrophobic patch in the SH3 domain (WK) alone reduce binding to PP196 and PP19C in co-IP experiments. Does a double mutation (FA+WK) compromise further the binding to PP1, thus revealing a synergistic or additive contribution of these regions for the interaction with PP1? This would correlate nicely with the *in vivo* data presented in Fig S9 and argue in favour of multiple motifs increasing the binding affinity for PP1.

We have included this experiment in the manuscript (see Figure 4D). As can be seen mutation of the two motifs completely abolishes the binding between the two proteins by co-IP. However, it should be noted that, as shown in our new pulldown/Mass Spectrometry experiment, mutation of the specificity pocket alone is already sufficient to dramatically reduce binding to PP1 (Figure 8A, S9), and functionally, mutation of either the RVxF motif or the SH3 specificity pocket alone is sufficient to severely reduce the ability of ASPP2 to induce TAZ dephosphorylation (Figure 8D, E).

2) Can the authors demonstrate *in vivo* to which extent is the interaction of PP1 with the ASPP-FA, WK and KVK mutants (alone and in combination with *ccdc85*^{-/-}) compromised? ASPP localizes at adherens junctions. Do these mutations preclude PP196A and/or PP19C localization at these structures? Is the dephosphorylation of ASPP-PP1 substrates affected in these mutants?

To directly link the ASPP/PP1 interaction with ASPP function, we have performed two new experiments:

- Using the known ASPP/PP1 substrate TAZ (a transcriptional co-activator target of the Hippo tumour-suppressor pathway), we show that, similar to mutations of the RVxF motif, the ASPP2^{KVK} mutation in the SH3 domain specificity pocket prevents ASPP2 from promoting TAZ dephosphorylation on Serine 89 in cell culture (Figure 8D, E). These data link our biophysical analysis with ASPP2 functions.
- Our previous biochemical data had shown that *Drosophila* ASPP interacts robustly with PP1α96A and PP1β9C, which have a C-tail, and only weakly with PP1α87C and PP1α13C, which lack a C-tail (Figure 1B). We examine the localization of PP1 isoforms *in vivo* in the *Drosophila* retina and identified a pool of PP1α96A and PP1β9C (also known as Flap wing – Flw) localized at the adherens junctions (Figure S10J-K'), where ASPP and RASSF8 are localized (Langton et al Curr Biol 2009), whereas PP1α87C shows no junctional enrichment (Figure S10I, I'). Interestingly, disrupting the ASPP/PP1 complex by loss of ASPP and Ccdc85 causes the loss of the junctional pools of PP1α96A and PP1β9C (Figures 9K-N and S10L-N). This shows that the ASPP complex is required to direct the subcellular localization of PP1 isoforms *in vivo*.

In addition, we have performed new pulldown/Mass Spectrometry experiments. We show that ASPP2 is strongly associated with PP1α and the ASPP2^{KVK} mutation, which disrupts the interaction between the ASPP2 SH3 domain and the PP1 C-tail *in vitro*, disrupts the association with PP1α without affecting other ASPP2 partners (Figure 8A, S9). In contrast, the weaker association of ASPP2 with PP1β and PP1γ1 is not

affected by the ASPP2^{KVK} mutation. This suggests that the ASPP2 SH3 domain specificity pocket is indeed required for PP1 α recruitment in a cellular context.

Minor points:

1) The authors claim in the abstract that PP1 association with *Drosophila* ASPP is required for ALL *in vivo* functions (line 39) of the latter. This feels like an overstatement. It should be stated that PP1 association is required for *Drosophila* ASPP functions under study.

As also requested by reviewer 1, we have changed this sentence to: “We show that *Drosophila* ASPP is part of a multiprotein PP1 complex and that PP1 association is necessary for several *in vivo* functions of *Drosophila* ASPP.”

2) The authors demonstrate in the present manuscript that *ccdc85* mutants phenocopy the ASPP mutant eye phenotype. This resembles the phenotype of RASSF8 mutants that had been previously described by the same lab, as referenced in the text. Although I agree with the conclusion, no RASSF8 mutants were analysed in the present study. It seems more accurate to report in the title of the RESULTS section “*ccdc85* mutants phenocopy the ASPP mutant eye phenotype” (line 187).

We have changed it in the text.

3) In the graphs of Fig 2F and 2G, GMR>*cc85* should be replaced by GMR>*ccdc85*.

This has now been corrected.

4) In the graphs of Fig 2L and 2M it should be explicit that the transgenes ASPP-WT and ASPP-FA are expressed in a ASPP-null background.

This has been done due to limited space in the Figure in order to keep the font size legible. The fact that the experiment is performed in an *ASPP* null background is mentioned in the main text, and in the figure legends, which also refer to the genotypes section of the manuscript.

5) The overlay in Figure 3L only displays the MS1096>ASPP-WT genotype. The MS1096>ASPP-FA seems to be missing.

We thank the reviewer for spotting this mistake, which has now been corrected.

Reviewers' Comments:

Reviewer #1:

Remarks to the Author:

The addition of new data and significant changes to the prior text have together greatly improved the manuscript.

Reviewer #2:

Remarks to the Author:

The revised version of the manuscript entitled "ASPP proteins discriminate between PP1 catalytic subunits through their SH3 domain and the PP1 C-tail" has significantly improved. The authors have addressed the specific questions and comments adequately. They added additional experimental data to support their hypothesis, improved the data analysis and integrated the structural and cell biology data. Overall, the different parts of the manuscript are better integrated and I do not have any additional comments or requests.

Reviewer #3:

Remarks to the Author:

The authors have thoroughly addressed my concerns. I am now convinced of the biological relevance of the selectivity. I think that this manuscript is of broad interest and recommend it for publication in Nature Communications.

Reviewer #4:

Remarks to the Author:

The authors have adequately addressed all my previous comments and the manuscript has been significantly improved. The reported findings are solid and the claims are well supported by the data. This manuscript contributes with new knowledge to better understand how specific PP1 complexes are formed and it is expected to be of interest to a broad audience. I recommend its publication in Nature communications.

REVIEWERS' COMMENTS:

Reviewer #1 (Remarks to the Author):

The addition of new data and significant changes to the prior text have together greatly improved the manuscript.

Reviewer #2 (Remarks to the Author):

The revised version of the manuscript entitled "ASPP proteins discriminate between PP1 catalytic subunits through their SH3 domain and the PP1 C-tail" has significantly improved. The authors have addressed the specific questions and comments adequately. They added additional experimental data to support their hypothesis, improved the data analysis and integrated the structural and cell biology data. Overall, the different parts of the manuscript are better integrated and I do not have any additional comments or requests.

Reviewer #3 (Remarks to the Author):

The authors have thoroughly addressed my concerns. I am now convinced of the biological relevance of the selectivity. I think that this manuscript is of broad interest and recommend it for publication in Nature Communications. Reviewer #4 (Remarks to the Author): The authors have adequately addressed all my previous comments and the manuscript has been significantly improved. The reported findings are solid and the claims are well supported by the data. This manuscript contributes with new knowledge to better understand how specific PP1 complexes are formed and it is expected to be of interest to a broad audience. I recommend its publication in Nature communications.